# Genome-wide CRISPR screen identifies noncanonical NF-κB signaling as a regulator of density-dependent proliferation

**Maria Fomicheva, Ian G Macara***

Department of Cell and Developmental Biology Vanderbilt University School of Medicine Nashville, Nashville, United States

**Abstract** Epithelial cells possess intrinsic mechanisms to maintain an appropriate cell density for normal tissue morphogenesis and homeostasis. Defects in such mechanisms likely contribute to hyperplasia and cancer initiation. To identify genes that regulate the density-dependent proliferation of murine mammary epithelial cells, we developed a fluorescence-activated cell sorting assay based on fluorescence ubiquitination cell cycle indicator, which marks different stages of the cell cycle with distinct fluorophores. Using this powerful assay, we performed a genome-wide CRISPR/Cas9 knockout screen, selecting for cells that proliferate normally at low density but continue to divide at high density. Unexpectedly, one top hit was *Traf3*, a negative regulator of NF-κB signaling that has never previously been linked to density-dependent proliferation. We demonstrate that loss of *Traf3* specifically activates noncanonical NF-κB signaling. This in turn triggers an innate immune response and drives cell division independently of known density-dependent proliferation mechanisms, including YAP/TAZ signaling and cyclin-dependent kinase inhibitors, by blocking entry into quiescence.

**\*For correspondence:**
ian.g.macara@vanderbilt.edu

**Competing interests:** The authors declare that no competing interests exist.

## Introduction

An important characteristic of epithelial cells is that, unlike fibroblasts, they do not undergo contact inhibition but continue to proliferate at confluence. This behavior enables the expansion of epithelial tissues during organismal growth without compromising the barrier function created by intercellular junctions. Importantly, however, proliferation is not indefinite but terminates at a preset cell density (*Fomicheva et al., 2020*). Stretching or wounding an epithelial sheet, which reduces the cell density, can reinitiate cell division (*Aragona et al., 2013*). Conversely, compression, which increases density, can result in extrusion and apoptosis of cells so as to bring the epithelial layer back to its homeostatic state (*Eisenhoffer et al., 2012*). This control mechanism that prevents tissue overgrowth is essential for normal development, and it is lost in hyperplasia and in cancer. However, the mechanisms that underlie homeostatic cell density maintenance remain incompletely understood.

One system through which epithelial cells can respond to changes in density is the Hippo pathway and its effectors YAP and TAZ. These transcriptional co-activators are nuclear at low cell density or under conditions of high mechanical strain, but become phosphorylated and are cytoplasmic (or junction-associated) and nonfunctional at high density and/or low strain (*Dupont et al., 2011*). Independently of YAP/TAZ, however, the polarity proteins LGL1/2 can also control density-dependent proliferation by inhibiting proteasomal degradation of the cyclin-dependent kinase inhibitor (CKI) CDKN1B (p27). Loss of LGL1/2 reduces the expression of p27, which is often upregulated at high cell density to arrest proliferation (*Yamashita et al., 2015*). Numerous studies on Hippo/YAP have demonstrated the complexity of this pathway, and several novel Hippo pathway components have

been revealed over the past few years. In addition, as mentioned previously, Hippo-independent mechanisms have also been recently reported, such as the LGL/p27 pathway. It is, therefore, likely that other Hippo pathway components and/or currently unknown Hippo-independent signaling mechanisms exist; however, no strategy has been designed to identify such mechanisms.

With the goal of discovering novel factors that regulate epithelial homeostasis, we developed a powerful new assay based on a fluorescence-activated cell sorting (FACS) approach that we integrated with a genome-wide CRISPR/Cas9 sgRNA knockout (KO) screen to select for genes that are essential for cell cycle arrest at high density, but which do not impact proliferation of cells below the threshold for arrest. The screen employs a fluorescence ubiquitination cell cycle indicator (FUCCI) system to mark proliferating cells in S/G2/M phases of cell cycle with a green fluorescent protein, and cells in G1/G0 with a red fluorescent protein (*Sladitschek and Neveu, 2015*). We found that mammary EpH4 epithelial cells robustly arrest in G1 or G0 by 4 days post-confluency. When these cells were transduced with a pooled whole-genome CRISPR KO library, and then sorted 4 days post-confluencyfor cycling cells, we identified several candidate genes that may regulate cell density-dependent proliferation activity. The top hit was *Nf2* (also called *Merlin*), a known tumor suppressor that negatively regulates YAP/TAZ, which validated our approach (*Petrilli and Fernández-Valle, 2016*). A second, unexpected hit was *Traf3*, a negative regulator of NF-κB signaling, which has never previously been reported to regulate cell density-dependent proliferation. We demonstrate that loss of *Traf3* robustly and specifically activates the noncanonical NF-κB pathway. This in turn triggers an innate immune response and cell autonomously drives cell division independently of both YAP/TAZ signaling and CKIs, overriding these classical mechanisms of density-dependent proliferation control and preventing cells at high density from entering quiescence.

## Results

### A FUCCI-based screen for density-dependent cell cycle arrest

Our goal was to design a screen for the rapid and efficient selection of epithelial cells that continue to proliferate inappropriately at high cell density. For the screen, we needed to identify a cell line that retained epithelial features, including homeostatic density control. We chose the murine EpH4 mammary epithelial cell line for this screen, because EpH4 cells are highly polarized, form confluent epithelial sheets, and, most importantly, we confirmed that they efficiently arrest at high density. We also needed a tool to specifically identify and select cells that maintain proliferative activity at high density. To distinguish cycling from non-cycling cells, we established a stable EpH4 line that expresses ES-FUCCI, which labels cells in G1/G0 with mCherry and cells in S/G2/M with mCitrine (*Figure 1A*; *Sladitschek and Neveu, 2015*). As expected, the EpH4-FUCCI cells remain proliferative at 1 day post-confluency but very few cells cycle at high density, with only about 1% of cells expressing mCitrine at 4 days post-confluency (*Figure 1B*, *Figure 1—figure supplement 1A*).

We reasoned that the expression of any shRNA or sgRNA that disrupts expression of a gene involved in density-dependent arrest could be identified by employing FACS to enrich for green cells from an EpH4-FUCCI population grown to high density. To test this concept, we used a shRNA lentiviral construct to deplete the CKI *p27*, which is known to be associated with cell cycle arrest post-confluency (*Yamashita et al., 2015*). Quantitative PCR for the *p27* transcript showed that ~60% of *p27* mRNA was lost in cells expressing the *p27* shRNA compared to EpH4-FUCCI cells expressing scrambled shRNA (*Figure 1—figure supplement 1B*). The p27-depleted cells display increased cell proliferation after 4 days post-confluency in contrast to cells expressing scrambled shRNA (*Figure 1—figure supplement 1A*). As a proof-of-principle experiment, we next mixed wild-type (WT) EpH4-FUCCI cells and sh-*p27* EpH4-FUCCI cells at a 10:1 ratio (*Figure 1—figure supplement 1C*). The mixed cells were then plated at a cell density of 100,000 cells/cm$^2$, such that they reached confluency 24 hr after seeding. We grew the mixture of cells for 4 days post-confluency and sorted mCitrine-positive (mCitrine+) cells by FACS. After sorting, we isolated genomic DNA (gDNA) and performed qPCR using primers against the puromycin gene, located in the lentiviral plasmid insert, to assess the fraction of p27-depleted cells in the cell mixture. We observed an average of 1.4× and 3.5× enrichment of the puromycin gene after the first and second rounds of sorting, respectively (*Figure 1—figure supplement 1D*). These data support the validity of our screen design.

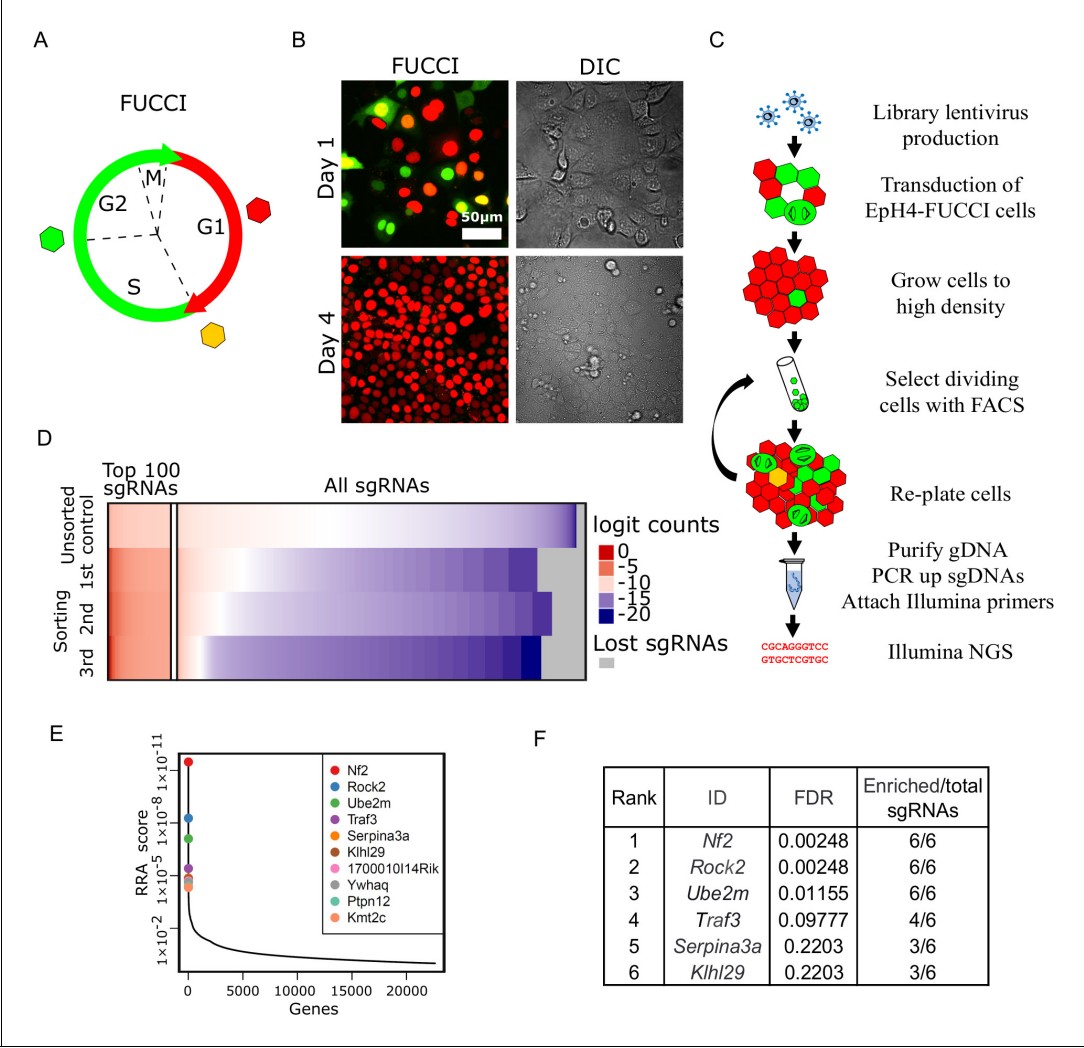

**Figure 1.** Whole-genome screening for genes that inhibit proliferation at homeostatic cell density. (A) Schematic of fluorescence ubiquitination cell cycle indicator (FUCCI) color transitions through cell cycle. (B) EpH4-FUCCI stable cell line grown to 1 and 4 days post-confluency. (C) Whole-genome CRISPR Knock Out screening strategy. (D) Read count distribution for samples before sorting and after different rounds of sorting. Data are logit transformed (f(p) = log$_2$(p/1 - p) where p is the proportion of a given sgRNA in the total number of sgRNAs in a sample). Color coding shows depleted sgRNAs in blue, enriched sgRNAs in red, and sgRNA with no enrichment in white. Gray shows lost sgRNAs. (E) Genes plotted based on their RRA enrichment score (third sorting). (F) List of genes with FDR below 0.25 and ≥3 sgRNAs enriched compared to control after third sort.

The online version of this article includes the following source data and figure supplement(s) for figure 1:

**Source data 1.** Source data file for *Figure 1*.
**Figure supplement 1.** Proof of principle experiments for the whole-genome screen.
**Figure supplement 1—source data 1.** Source data file for *Figure 1—figure supplement 1*.
**Figure supplement 2.** sgRNA sequence PCR for next-generation sequencing (NGS) and sequence processing.

Based on these encouraging results, we proceeded to developing an EpH4-FUCCI cell line, in which every cell has lost a single gene. We transduced EpH4-FUCCI cells with the pooled GeCKO CRISPR v2 KO lentivirus library (*Shalem et al., 2014*; *Sanjana et al., 2014*; *Figure 1D*). The library contains six sgRNAs against each of 20,611 genes, plus 1000 non-targeting controls, for a total of 130,209 sgRNAs. To ensure that our EpH4-FUCCI cells would receive only one sgRNA on average, we transduced cells with viruses at a multiplicity of infection (MOI) of 0.3. It is critical to maintain a large enough cell population to ensure that all sgRNAs in the library are retained. Therefore, we aimed to have 150–300 cells per sgRNA in each step of the screen. Based on these values, 6.7 × 10$^7$ EpH4-FUCCI cells were transduced to obtain about 2 × 10$^7$ cells that had acquired viruses (~150

cells per sgRNA). Cells that were not infected with virus were eliminated by puromycin selection. We plated the cells at 100,000 cells/cm$^2$ and grew them for 4 days post-confluency. To maintain library representation, we sorted $4 \times 10^7$ cells (300 cells per sgRNA) by FACS to select mCitrine+ cells. We re-grew cells for the next round of selection and for gDNA isolation. In total, we performed three FAC sorts. We observed a steady increase in the number of mCitrine+ cells after each round of FACS (*Figure 1—figure supplement 1E and F*). gDNA was purified, and integrated sgRNA cassettes were amplified from the DNA by PCR, as described previously (*Shalem et al., 2014*; *Sanjana et al., 2014*; *Figure 1—figure supplement 2A and B*). A second PCR reaction attached Illumina index primers (*Figure 1—figure supplement 2A and C*), and the pooled product was then sequenced.

We used the MAGeCK program (*Li et al., 2014*) to map sequencing reads to the library (*Figure 1E*, *Figure 1—figure supplement 2D and D'*). Read count distribution was relatively uniform in the control (unsorted) sample, with depletion of those sgRNAs that target essential genes. After each round of selection, more sgRNAs were depleted, but a small portion of sgRNAs was highly enriched. We utilized the MAGeCK algorithm to find genes that were enriched after sorts compared to control. MAGeCK takes into account changes in the abundance of all sgRNAs targeting a single gene, measured by the robust ranking aggregation (RRA) score. The most enriched genes have the smallest RRA scores (*Figure 1F*, *Figure 1—figure supplement 2E and E'*). We selected genes with FDR below 0.25 and with three or more sgRNAs/gene selected (*Figure 1G*, *Figure 1—figure supplement 2F and F'*).

## The CRISPR screen identifies a component of the Hippo pathway, a lymphocyte proliferation control factor, and NEDD8-conjugating E2 enzyme

Interestingly, the top enriched target was *Nf2/Merlin*, which is a known tumor suppressor gene and a regulator of Hippo signaling necessary for cell density control (*Petrilli and Fernández-Valle, 2016*). This hit, the sgRNAs for which were enriched by >1000× above the control abundance, strongly validated our screening strategy. Two unexpected hits were for the *Traf3* and *Ube2m (Ubc12)* genes (*Figure 1G*), which have not previously been implicated in density-dependent cell cycle arrest.

TRAF3 (TNF receptor associated factor 3) is critical for lymphocyte proliferation control and immune responses (*Zapata et al., 2009*). It negatively regulates signaling through multiple pathways, including the canonical and noncanonical NF-κB pathways (*He et al., 2006*; *Ramakrishnan et al., 2004*; *Zarnegar et al., 2008*; *Sun, 2017*).

Deletions and mutations of *TRAF3* are among the most common genetic alterations in human B cell malignancies (*Zhu et al., 2018*). B cell-specific KO of the *Traf3* gene in mice leads to increased B cell numbers and spontaneous lymphomas (*Moore et al., 2012*). Myeloid-specific *Traf3* KO causes histiocytic sarcomas of macrophage origin (*Lalani et al., 2015*). Therefore, TRAF3 is critical for the prevention of malignant growth in blood cells; however, TRAF3 roles in epithelia have not yet been widely investigated, despite its ubiquitous expression in mouse and human tissues (*Yue et al., 2014*).

A third hit from our screen was *Ube2m* (ubiquitin conjugating enzyme E2 M, also known as *Ubc12*). UBE2M is a NEDD8 conjugation E2 enzyme, which neddylates CULLIN-RING ligases to stimulate their activity (*Lydeard et al., 2013*). Interestingly, stress induces UBE2M expression and promotes its ubiquitylation of UBE2F, the degradation of which can suppress cell proliferation (*Zhou et al., 2018*).

To confirm that loss of our top candidates indeed results in over-proliferation at high density, we developed two new sgRNA CRISPR v2 plasmids per gene. The sgRNA sequences were different from those in the GeCKO library. We transduced EpH4 cells with these sgRNA lentiviruses and confirmed efficient KO of each gene by immunoblotting for NF2, TRAF3, and UBE2M (*Figure 2A–C'*). To test if loss of the target genes leads to a failure in cell cycle arrest at high density, we analyzed BrdU incorporation at 1 and 4 days post-confluency. BrdU was added to cells for 1 hr, then cells were fixed and stained. When cells had just reached a confluent state, we found that BrdU incorporation was similar for the non-targeted sgRNA control (NT) and each of the KO cell lines, but that at high density the KO cells incorporated more BrdU than did the NT control cells (*Figure 2D*). These differences were statistically significant at high density, as determined by cytometric analysis of BrdU

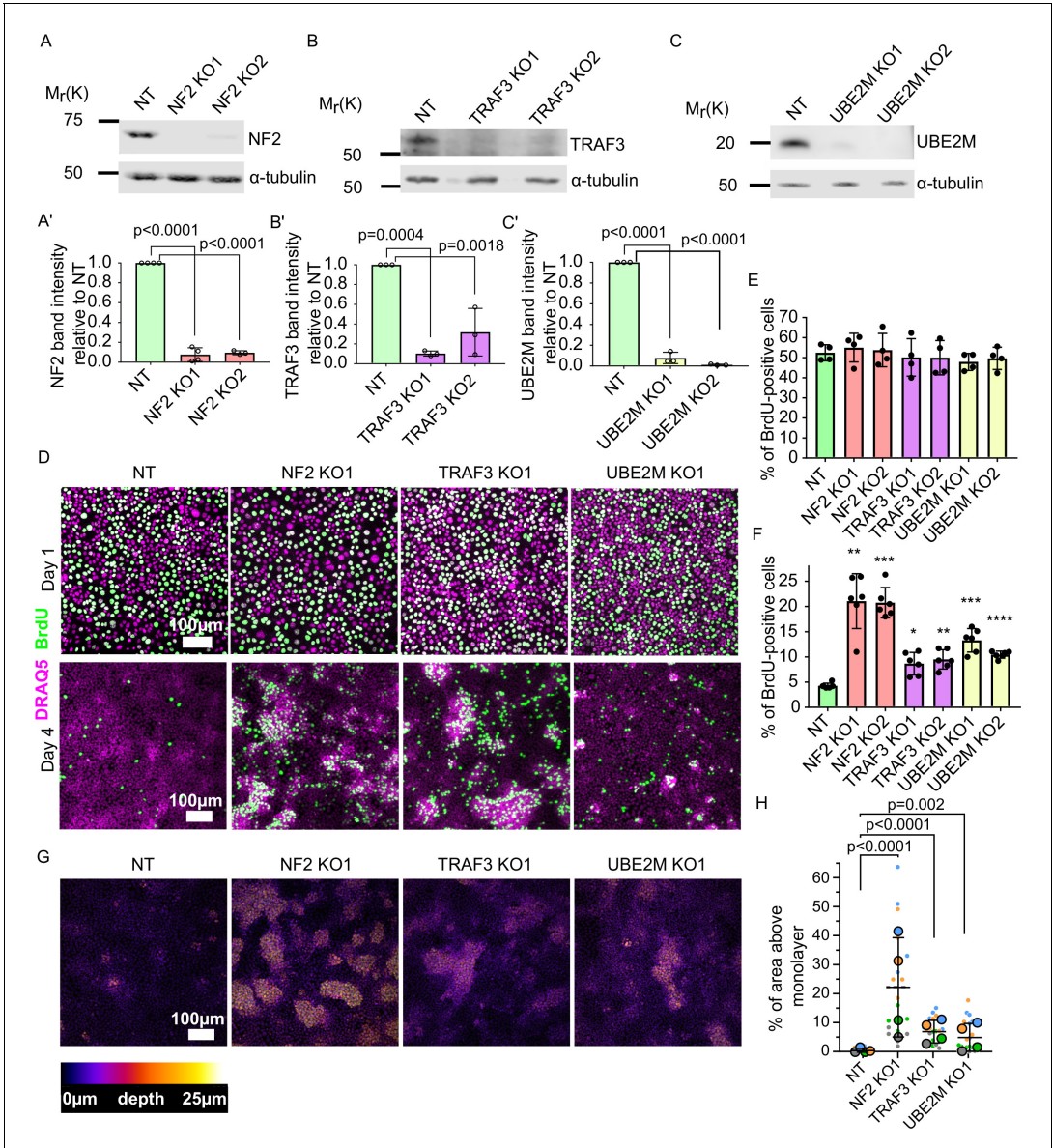

**Figure 2.** Validation of candidate genes identified from the screen. (**A–C**) Immunoblots of NT control and NF2 (**A**), TRAF3 (**B**), and UBE2M (**C**) KO EpH4 cells for NF2, TRAF3, and UBE2M, respectively. α-tubulin or GAPDH was used as a loading control. (**A'–C'**) Quantification of NF2 (**A'**), TRAF3 (**B'**), and UBE2M (**C'**) levels based on immunoblotting in (**A–C**). (**A'C'**) Histograms show mean ± 1 s.d. (n = 3). p-values were calculated by one-way ANOVA followed by Dunnett's multiple comparisons test. (**D**) NT control, NF2, TRAF3, and UBE2M KO cells grown for 1 or 4 days post-confluency. BrdU was added for 1 hr, and then cells were fixed and stained for BrdU, with DAPI as a nuclear marker. (**E and F**) Cytometric analysis of NT control, NF2, TRAF3, and UBE2M KO cells stained for BrdU to assess proliferation at 1 or 4 DPC, respectively. n = 4 (**E**) and n = 6 (**F**), histograms show mean ± 1 s.d. p-values were calculated by one-way ANOVA followed by Dunnett's multiple comparisons test. (**G**) Control and KO cells stained with Hoechst dye. Confocal images are depth color coded. The color code scale is shown below. (**H**) Five fields of view in four biological repeats were used to quantify the level of multilayering. Data shown as a SuperPlot (***Lord et al., 2020***). p-values were calculated by mixed model two-way ANOVA.

The online version of this article includes the following source data and figure supplement(s) for figure 2:

**Source data 1.** Source data file for ***Figure 2***.
**Figure supplement 1.** BrdU+ cell cytometry gates and cancer survival based on *Traf3* expression level.

+ cells (***Figure 2E and F***, ***Figure 2—figure supplement 1A***). We also observed that all three KO cell lines formed multiple layers at high density, while control cells remained as a single, uniform layer (***Figure 2G and H***, ***Figure 2—figure supplement 1B***). Together, these data demonstrate that our

screen successfully identified genes that are essential for the restriction of proliferation at high cell density. Importantly, loss of these genes has no effect on the cell cycle at lower densities.

## TRAF3 suppresses proliferation of mammary organoids, human mammary epithelial cells, and fibroblasts

Little is known about the function of TRAF3 in epithelial cells, and its role in density-dependent cell cycle arrest has not been previously investigated. Interestingly, however, genetic alterations in *Traf3* occur in a variety of human epithelial cancers, though at a level of <6% (*Zhu et al., 2018*). Low *Traf3* mRNA expression is also associated with significantly worse survival for lung and gastric cancer patients (*Figure 2—figure supplement 1C*; *Nagy et al., 2018*). Therefore, we focused on this gene for further analysis.

To determine the generality of the phenotype induced by deletion of *Traf3*, we first asked if the effects are confined to the EpH4 mammary epithelial line or are also important in primary mammary tissue. To address this question, we used murine mammary organoids, which recapitulate many aspects of normal morphogenesis of the mammary gland (*Pasic et al., 2011*; *Ewald et al., 2008*). Mammary gland ductal fragments were isolated from WT C3H mice, transduced with lentivirus, and then grown as organoids in Matrigel culture, as described previously (*Pasic et al., 2011*; *Figure 3A*). As shown in *Figure 3B*, WT organoids form buds with hollow lumens, but organoids lacking TRAF3 formed multilayered buds with small or no detectable lumens. Additionally, staining for phospho-HISTONE H3 revealed a substantially higher mitotic index in organoids deleted for *Traf3*, compared to organoids transduced with a control sgRNA (*Figure 3B and C*), demonstrating that TRAF3 normally suppresses WT tissue overgrowth.

To extend our analysis across species, we transduced the normal human mammary gland cell line MCF10A with sgRNAs designed to target human *Traf3*. Loss of the gene product was confirmed by immunoblotting (*Figure 3—figure supplement 1A and A'*). We assessed cell proliferation by immunofluorescence and cytometric analysis of BrdU incorporation as described above, and found that,

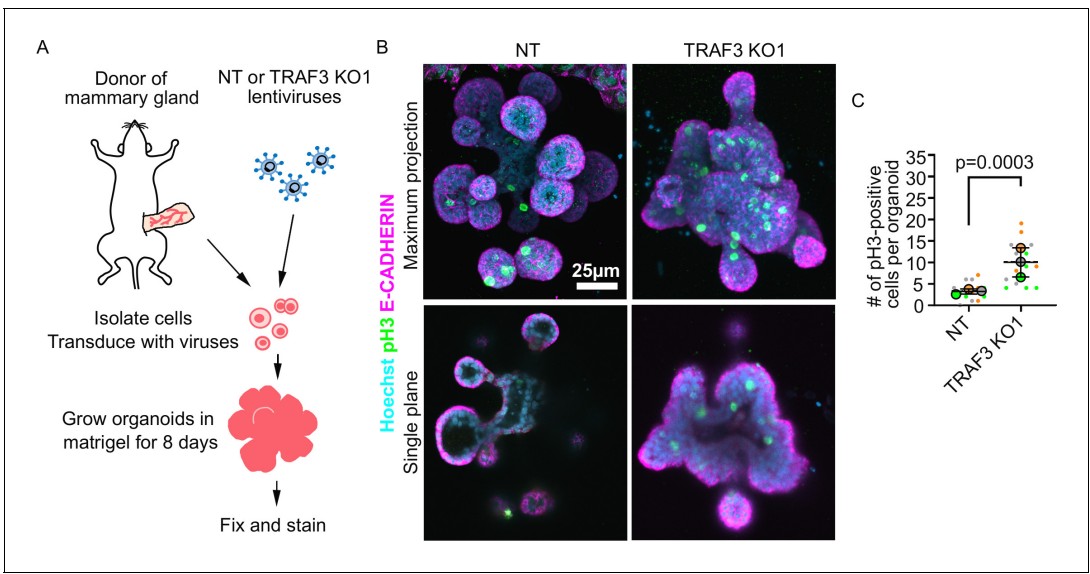

**Figure 3.** Loss of TRAF3 causes over-proliferation in primary mammary organoids. (**A**) Primary mammary organoid experiment workflow. (**B**) Maximum intensity projection of NT and TRAF3 KO1 mammary organoids stained for phospho-HISTONE H3 (pH3), E-CADHERIN, and DNA (Hoechst). Bottom panel – single confocal plane of NT and TRAF3 KO1 organoids. (**C**) Quantifications of the number of pH3-positive cells per organoid; 5–10 organoids of comparable size were quantified per condition per repeat. Data shown as a SuperPlot (n = 3). p-values were calculated by mixed model two-way ANOVA.

The online version of this article includes the following source data and figure supplement(s) for figure 3:

**Source data 1.** Source data file for *Figure 3*.

**Figure supplement 1.** Loss of TRAF3 in MCF10a and NIH 3T3 cells causes over-proliferation.

**Figure supplement 1—source data 1.** Source data file for *Figure 3—figure supplement 1*.

similar to EpH4 cells, MCF10a TRAF3 KO cells, but not WT MCF10a cells, over-proliferate at high density (*Figure 3—figure supplement 1B and B'*).

Finally, we determined if this role for TRAF3 is confined to epithelial cells by knocking out the gene in NIH 3T3 fibroblasts. Normally, these cells contact-inhibit at high density. CRISPR KO of *Traf3* (confirmed by western blotting (*Figure 3—figure supplement 1D and D'*)) promoted cell proliferation at high density as measured by cytometric analysis of BrdU staining (*Figure 3—figure supplement 1D and D'*). These data suggest that loss of TRAF3 broadly interferes with cell cycle arrest including cells of epithelial and mesenchymal lineages.

## Loss of TRAF3 activates noncanonical NF-κB signaling to promote over-proliferation

How does TRAF3 regulate proliferation? We hypothesized that TRAF3 might be involved in the same pathways as in blood cells, where it negatively regulates both the canonical and noncanonical NF-κB pathways (*He et al., 2006*; *Ramakrishnan et al., 2004*; *Zarnegar et al., 2008*; *Sun, 2017*). In the noncanonical pathway, TRAF3 constitutively promotes proteasomal degradation of MAP3K14 (NIK), a kinase that is essential for activation of the downstream signaling cascade. Ligand recruitment to an upstream receptor triggers TRAF3 poly-ubiquitinylation and degradation. As a result, NIK levels increase, and it phosphorylates its downstream target IKKα, which in turn phosphorylates NFKB2 (p100). Phosphorylated p100 is partially degraded to p52, which enters the nucleus in association with RELB and regulates the transcription of target genes (*Sun, 2017*).

To test if loss of TRAF3 activates noncanonical NF-κB signaling, we examined the location of RELB in control and TRAF3 KO EpH4 cells and primary mammary gland organoids (*Figure 4A–C*). As expected, RELB was predominantly cytoplasmic in control cells and organoids. However, loss of TRAF3 caused a substantial nuclear accumulation, indicative of noncanonical NF-κB activation. We also performed immunoblotting for noncanonical NF-κB signaling components in control and TRAF3 KO cells at 1 and 4 days post-confluency (*Figure 4D and E*). TRAF3 KO cells at both 1 and 4 days post-confluency had substantially increased NIK levels and enhanced processing of p100 to p52 compared to NT control cells. However, cell density differences had no impact on NIK, RELB, TRAF3, and p100/p52 protein levels. Therefore, TRAF3 KO leads to activation of noncanonical NF-κB signaling, but regulation of these pathway components is independent of cell density.

Based on previous literature, we predicted that loss of TRAF3 could also stimulate canonical NF-κB signaling (*Figure 4F*). IκB normally blocks RELA/p65 and p50 from entering the nucleus, but activation of upstream kinases IKKα/β by phosphorylation induces IκB phosphorylation and degradation (*Ramakrishnan et al., 2004*; *Zarnegar et al., 2008*). Surprisingly, however, RELA/p65 did not accumulate in the nuclei of TRAF3 KO cells (*Figure 4G*). Immunoblotting for the canonical NF-κB pathway components showed activation of IKKα/β phosphorylation at high density in both control and TRAF3 KO cells, but no IκB degradation was observed in control and TRAF3 KO cells either at low or at high density (*Figure 4H*). Together, these data demonstrate that in mammary epithelial cells, the loss of TRAF3 specifically activates noncanonical but not canonical NF-κB signaling.

If TRAF3 affects cell proliferation via the noncanonical NF-κB pathway, then deletion of p100 in TRAF3 KO cells should reduce proliferation to control levels (*Figure 5A*). To test this prediction, we generated a lentivector encoding a p100 sgRNA, plus GFP as a selection marker. We transduced EpH4 cells with virus and isolated GFP+ cells by FACS. Immunoblotting confirmed efficient KO of p100 in NT control and TRAF3 KO cells (*Figure 5B and B'*). Blotting also showed increased cleavage of p100 to p52 in TRAF3 KO cells as compared to control cells (*Figure 5B*), and both bands were lost in cells expressing the *p100* sgRNA, confirming that the detected 52 kDa band is indeed the product of p100. We next grew cells to high density and analyzed pulsed BrdU incorporation by flow cytometry. There was no significant difference between control and NT/p100 KO cells, indicating that TRAF3 does not regulate cell cycling at low cell density. The proliferation of TRAF3/p100 double KO cells was, however, substantially reduced compared to TRAF3 KO cells (*Figure 5C and D*) demonstrating that noncanonical NF-κB signaling is required for the phenotype induced by loss of TRAF3.

As a further test, we knocked out the effector kinase NIK. Immunoblotting (*Figure 5E and E'*) revealed that, as expected, NIK is undetectable in NT control cells, but accumulates in TRAF3 KO cells. However, expression was lost in TRAF3/NIK double KO cells. Similar to p100 KO, deletion of NIK caused reduced proliferation in TRAF3 KO cells (*Figure 5F and G*).

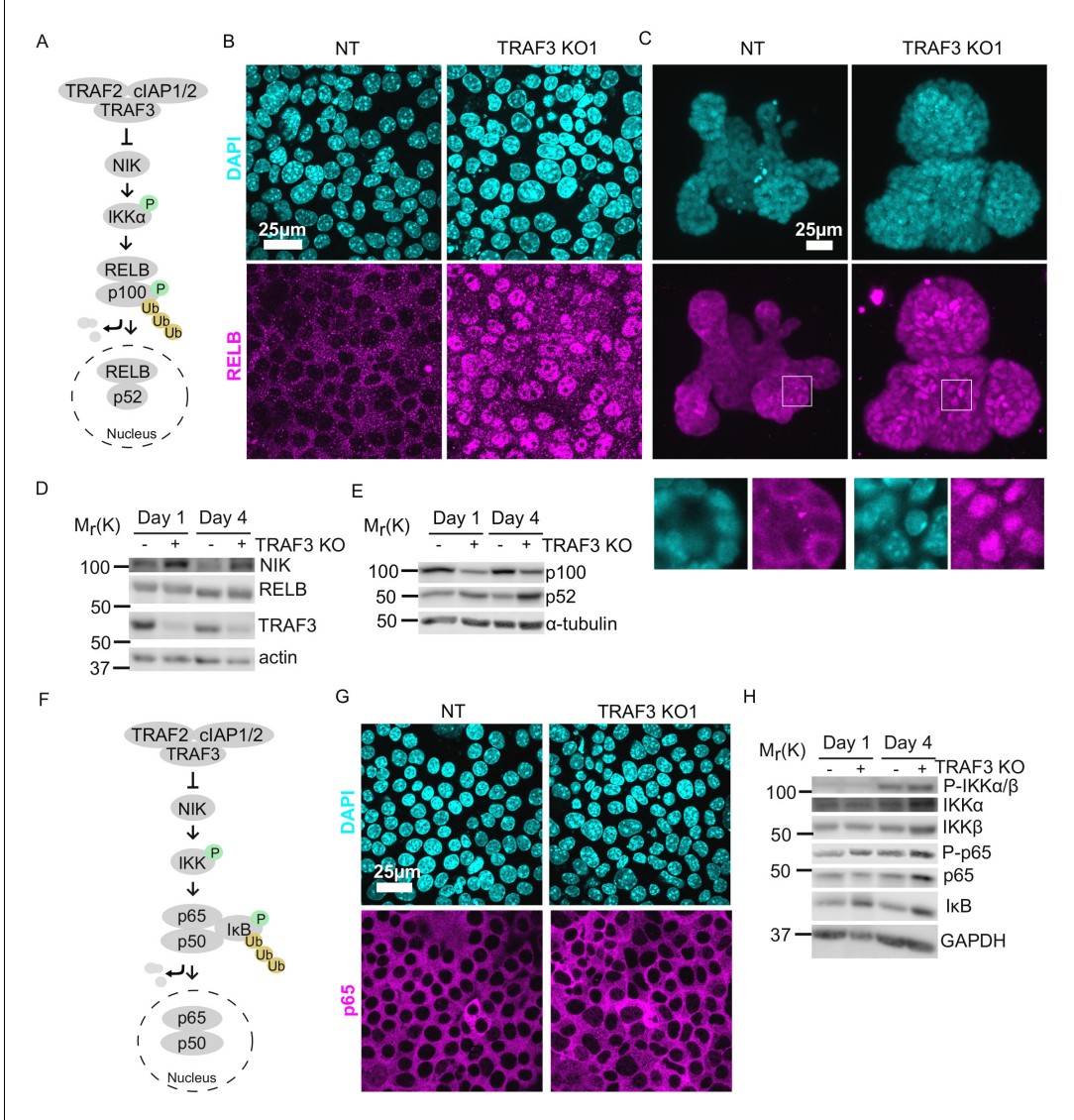

**Figure 4.** Loss of TRAF3 specifically activates noncanonical but not canonical NF-κB pathway. (**A**) Simplified schematic of noncanonical NF-κB pathway. TRAF3 sends NIK for degradation. In the absence of TRAF3, NIK levels increase, and it activates the downstream cascade, resulting in increased p100 processing to p52. RELB/p52 enters the nucleus where it regulates transcription of target genes. In experiments (**B**) and (**C**) RELB localization was tested in NT and TRAF3 KO1 cells. (**B and C**) Fluorescence staining of NT and TRAF3 KO1 EpH4 cells (**B**) and organoids (**C**) with anti-RELB antibodies and DAPI. (**B**) Single confocal planes. (**C**) Maximum intensity projection. White boxed ROIs are shown in enlarged images as single confocal planes. (**D**) Immunoblots of NT and TRAF3 KO1 lysates grown to 1 or 4 days post-confluency probed for NIK, RELB, TRAF3, and actin as loading control. (**E**) Immunoblots of NT and TRAF3 KO1 lysates grown to 1 or 4 days post-confluency probed for p100/p52 and α-tubulin as loading control. (**F**) A schematic of TRAF3-dependent activation of canonical NF-κB pathway. (**G**) Fluorescence staining of NT and TRAF3 KO1 cells with antibodies against p65 and with DAPI. (**H**) Western blotting of NT and TRAF3 KO1 cells at 1 and 4 days post-confluency probed for components of canonical NF-κB pathway and GAPDH as loading control.

To further confirm that noncanonical NF-κB pathway is necessary for the high density over-proliferation phenotype, we treated cells with BV6, a selective inhibitor of IAP proteins. cIAP1/2 is a part of the TRAF3 complex that degrades NIK. Therefore, we predicted that treatment with BV6 would induce over-proliferation. We treated cells with BV6 at low and high density (*Figure 5H and I*, respectively) and observed that BV6 treatment induces over-proliferation of NT cells at high density but not at low density, as shown by cytometric analysis of BrdU staining (*Figure 5H and I*).

We next asked if the noncanonical NF-κB pathway is sufficient to induce over-proliferation. We cloned and overexpressed a mutant NIK protein lacking the TRAF3-binding motif (NIK-ΔT3)

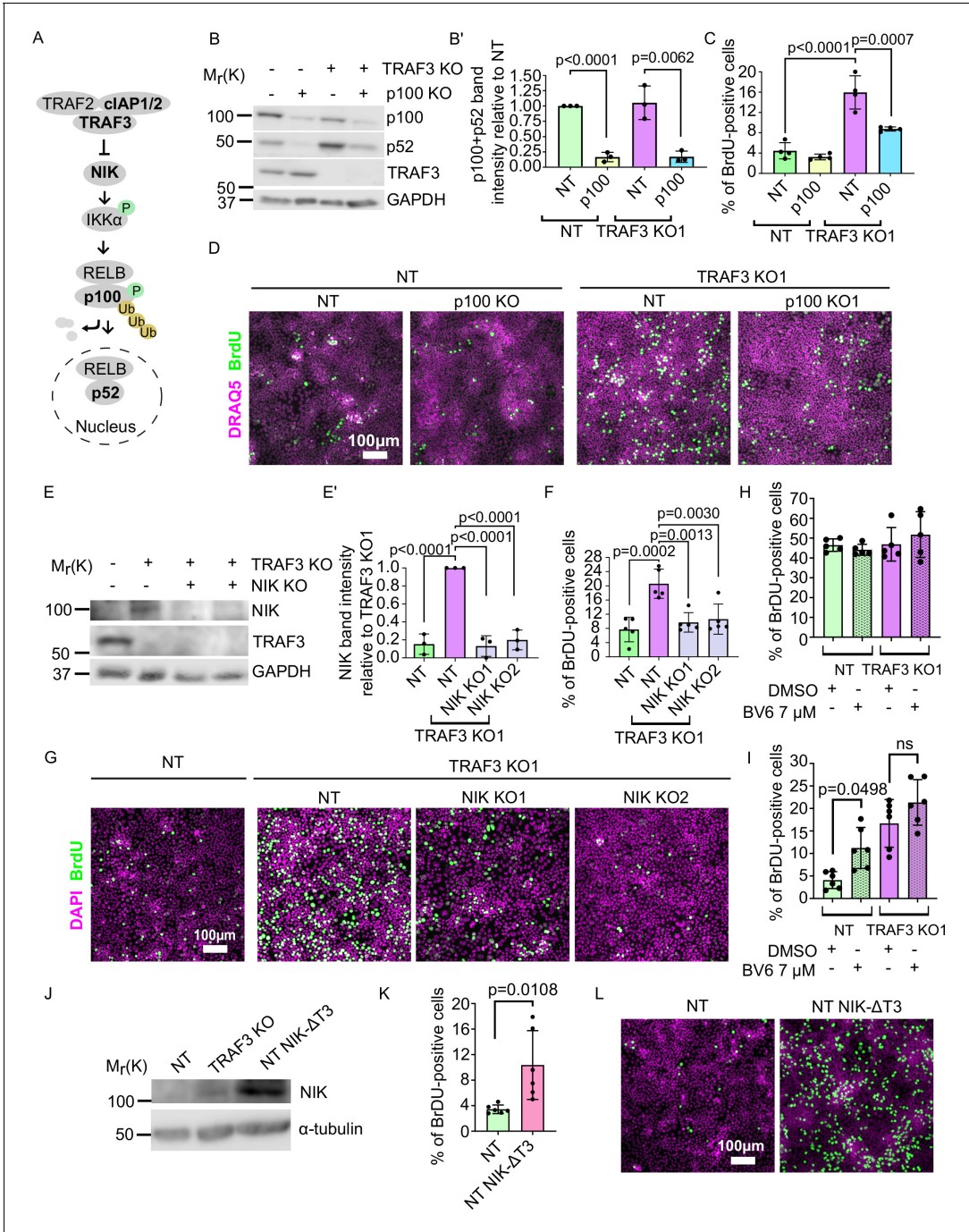

**Figure 5.** Noncanonical NF-κB signaling is necessary and sufficient for over-proliferation in TRAF3 KO cells. (**A**) Noncanonical NF-κB pathway schematic. Proteins shown in bold were manipulated in this study. (**B**) Immunoblotting of NT and TRAF3 KO1 cells transduced with pLVTHM GFP NT or *p100* sgRNAs. Blot was stained for p100/p52, TRAF3, and GAPDH (loading control). (**B′**) Quantifications of western blot (**B**). Histogram shows mean ± 1 s.d. (n = 3). p-values were calculated by Student's t-test. (**C**) Cytometric analysis of NT and TRAF3 KO cells transduced with pLVTHM GFP NT or *p100* sgRNAs. Graph shows mean ± s.d. (n = 4). p-values were calculated by one-way ANOVA followed by Tukey's multiple comparisons test. (**D**) NT and TRAF3 KO1 cells transduced with pLVTHM GFP NT or *p100* sgRNAs were treated with BrdU at 4 days post-confluency and immunostained for BrdU and DAPI. (**E**) Immunoblot of NT and TRAF3 KO1 cells transduced with NT, NIK KO1 or KO2. Blot was stained for NIK, TRAF3, and GAPDH (loading control). (**E′**) Quantifications of western blot (**E**). Histogram shows mean ± 1 s.d (n = 3). p-values were calculated by one-way ANOVA followed by Tukey's multiple comparisons test. (**F**) Cytometric analysis of NT cells and TRAF3 KO1 cells transduced with control NT, NIK KO1 and KO2. Histogram shows mean ± 1 s.d. (n = 5). p-values were calculated by one-way ANOVA followed by Tukey's multiple comparisons test. (**G**) NT and TRAF3 KO1 cells transduced with control NT, NIK KO1 and KO2 were treated with BrdU at 4 days post-confluency and immunostained for BrdU and DAPI. (**H and I**) Cytometric analysis of NT and TRAF3 KO cells treated with 7 μM BV6 at low (**H**) and high (**I**) density. Histogram shows mean ± 1 s.d. (n = 5 (**H**), n = 6 (**I**)).
*Figure 5 continued on next page*

*Figure 5 continued*

p-values were calculated by one-way ANOVA followed by Dunnett's (H) or Tukey's (I) multiple comparisons test. (J) Immunoblot of NT, TRAF3 KO1 (positive control), and NT cells expressing pWPI mScarlet NIK-ΔT3. Blot was stained for NIK and GAPDH (loading control). (K) Cytometric analysis of NT cells and NT pWPI mScarlet NIK-ΔT3 cells. Histogram shows mean ± 1 s.d. (n = 6). p-values were calculated by Student's t-test. (L) NT and NT pWPI mScarlet NIK-ΔT3 cells were treated with BrdU at 4 days post-confluency and immunostained for BrdU and DAPI.

The online version of this article includes the following source data for figure 5:

**Source data 1.** Source data file for *Figure 5*.

(*Liao et al., 2004*) to prevent NIK degradation (*Figure 5J*). Flow cytometry and immunofluorescent staining showed strong over-proliferation of NT NIK-ΔT3 cells at 4 days post-confluency compared to NT control cells (*Figure 5K and L*) demonstrating that activation of noncanonical NF-κB pathway is sufficient for increased proliferation at high density.

We conclude that TRAF3 normally suppresses proliferation at high cell density via inhibition of the noncanonical NF-κB pathway but does not impact proliferation at lower cell density. This pathway does not contribute to proliferation in WT EpH4 cells but is constitutively activated by loss of TRAF3. Activation of noncanonical NF-κB pathway downstream of TRAF3 is both necessary and sufficient for over-proliferation phenotype at high density.

## Noncanonical NF-κB signaling activates an innate immune response

Our data show that the loss of TRAF3 results in increased levels of the transcription factor p52, which is necessary for over-proliferation. However, gene expression changes downstream of noncanonical NF-κB signaling have not been investigated in epithelia, nor is it known which genes are targets of p52 in epithelial cells. Therefore, to identify downstream effectors of TRAF3 and noncanonical NF-κB signaling, we performed genome-wide transcriptomics analysis on NT control cells, TRAF3 KO, and TRAF3/p100 double KO cells. For these studies, we used our EpH4-FUCCI mammary epithelial cell line, which was transduced with NT, *Traf3*, or *Traf3* plus *p100* sgRNA lentiviruses, and sorted for G1/G0 phase (mCherry+) cells (*Figure 6A*). This strategy eliminates indirect gene expression differences reflective of the larger fraction of cycling (mCitrine+) cells caused by loss of TRAF3 but will not remove constitutive changes in gene expression.

Two biological replicates of each cell line were processed, and fastq files were analyzed using CLC Genomics Workbench and Ingenuity Pathway Analysis (IPA). Differential gene expression was filtered for greater than twofold differences, FDR < 0.05. Surprisingly, IPA did not detect any stimulation by TRAF3 KO of those signaling pathways classically involved in proliferation, including PI3K/AKT, JNK, TGF-β, WNT, and Hippo (*Figure 6B*). There was, however, a dramatic induction of innate immune response genes. Antigen presentation, interferon response genes, and genes responsible for foreign DNA and RNA recognition were strongly upregulated by deletion of TRAF3 (*Figure 6B–E*). Additionally, TRAF3 KO cells overexpress adhesion molecules *Icam1* and *Vcam1* that facilitate adhesion of recruited leukocytes (*Figure 6F*). Superoxide-producing NADPH-oxidase complex genes, which eliminate bacteria and fungi, were also induced (*Figure 6G*), as were several chemokines and growth factors (*Figure 6H*), but some genes that normally respond to pathogens, such as defensins, IL-8, and IL-17, were not affected.

The comparison of TRAF3 KO and TRAF3/p100 double KO samples revealed that not all pathways activated by TRAF3 KO could be reversed by blocking noncanonical NF-κB signaling. We found that p100 KO partially reduces antigen presentation gene expression, the interferon response pathway, and foreign DNA and RNA recognition genes (*Figure 6B–E*), but differential gene expression of other pathways was not significantly changed. These data argue that antigen presentation, interferon pathways, and foreign DNA and RNA recognition genes are downstream targets of noncanonical NF-κB signaling in mammary epithelial cells. However, none of these pathways is likely to promote cell proliferation. Rather, the data suggest that noncanonical NF-κB signaling instead suppresses signals that normally would cause cell cycle exit.

## Loss of TRAF3 does not affect YAP signaling

Hippo/YAP signaling is a key mechanism that prevents abnormal proliferation at high density. Therefore, we tested whether the response to deletion of TRAF3 is independent of this pathway. The

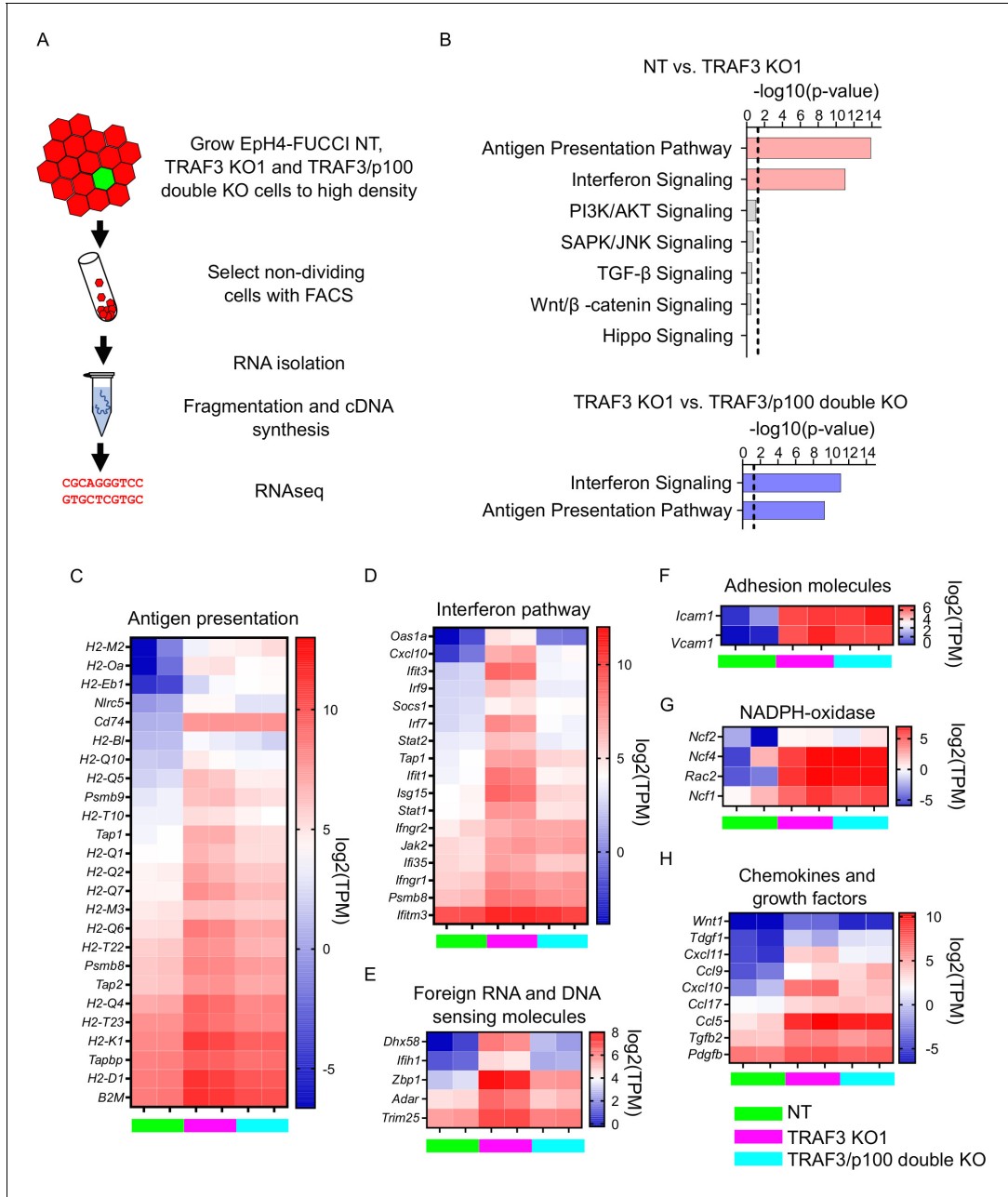

**Figure 6.** Loss of TRAF3 in EpH4 cells induces immune response pathways. (**A**) RNAseq experiment workflow. (**B**) Ingenuity Pathway Analysis (IPA) of RNAseq data showing significant upregulation of pathways related to infection response in TRAF3 KO cells compared to NT control (top, pink bars), and significant reduction of these pathways in TRAF3/p100 double KO cells compared to TRAF3 KO cells (bottom, blue bars). Notably, Hippo signaling and other pathways that regulate proliferation were unchanged. Threshold line is p<0.05. (**C–H**) Heatmaps comparing mRNA levels of antigen presentation (**C**), interferon (**D**), foreign RNA and DNA recognition (**E**) pathway components, adhesion molecules (**F**), NADPH-oxidase components (**G**), and chemokines and growth factors (**H**) between NT, TRAF3 KO1, and TRAF3/p100 double KO in two experimental repeats. NT, TRAF3 KO1, and TRAF3/p100 double KO are color coded with green, purple, and light blue, respectively. Data are presented as log2 of transcripts per million (TPM). The online version of this article includes the following source data for figure 6:

**Source data 1.** Source data file for *Figure 6*.

transcriptional co-activator YAP, a downstream effector of Hippo, is nuclear at low cell density and promotes cell cycling, but becomes phosphorylated and is cytoplasmic at high density (*Ma et al., 2019*). Therefore, we asked if loss of TRAF3 induces YAP translocation to the nucleus. We immunostained control, NF2 KO cells (positive control) and TRAF3 KO cells with anti-YAP antibodies. In

dense cultures, YAP was predominantly cytoplasmic in the NT control cells, but nuclear in NF2 KO cells, as expected. However, deletion of TRAF3 did not induce significant nuclear accumulation of YAP, as compared to the NT control (*Figure 7A and B*). Moreover, comparison of expression profiles for conserved YAP target genes (*Cordenonsi et al., 2011*) from our RNAseq data showed an absence of any significant induction for more than 90% of conserved YAP target genes in TRAF3 KO cells compared to the NT control (*Figure 7C*). We conclude, therefore, that TRAF3 loss promotes over-proliferation independently of YAP, a major regulator of cell density-dependent proliferation.

## The TRAF3 KO cell over-proliferation phenotype is cell autonomous

Our RNAseq data showed that several secreted factors including chemokines and growth factors (*Figure 6H*) are induced by loss of TRAF3, and we hypothesized that these factors might be secreted and trigger over-proliferation at high density in neighboring cells. To test this possibility, we labeled NT control cells with GFP and, separately, control or TRAF3 KO cells with mScarlet, then mixed red and green cells (NT + TRAF3 KO, or NT + NT) at a 1:1 ratio, and after 4 days post-confluency stained them for the proliferation marker KI-67 and DAPI (*Figure 7—figure supplement 1A*). Cytokines released from the TRAF3 KO cells could then, in principle, stimulate proliferation of surrounding NT (GFP+) control cells. However, there was no significant difference in the fraction of KI-67+ cells between NT GFP cells mixed with NT mScarlet and NT GFP cells mixed with TRAF3 KO mScarlet cells (*Figure 7—figure supplement 1B*). These data strongly argue that the over-proliferation phenotype is cell autonomous and is not a consequence of secreted cytokines or other factors.

## Loss of TRAF3 does not affect the levels of CKIs

The p27 CKI has been reported to inhibit density-dependent proliferation in response to changes in LGL1/2 expression (*Yamashita et al., 2015*). Moreover, we demonstrated that shRNA knockdown of *p27* can promote cell proliferation in EpH4 cells (*Figure 1B and C*, *Figure 1—figure supplement*

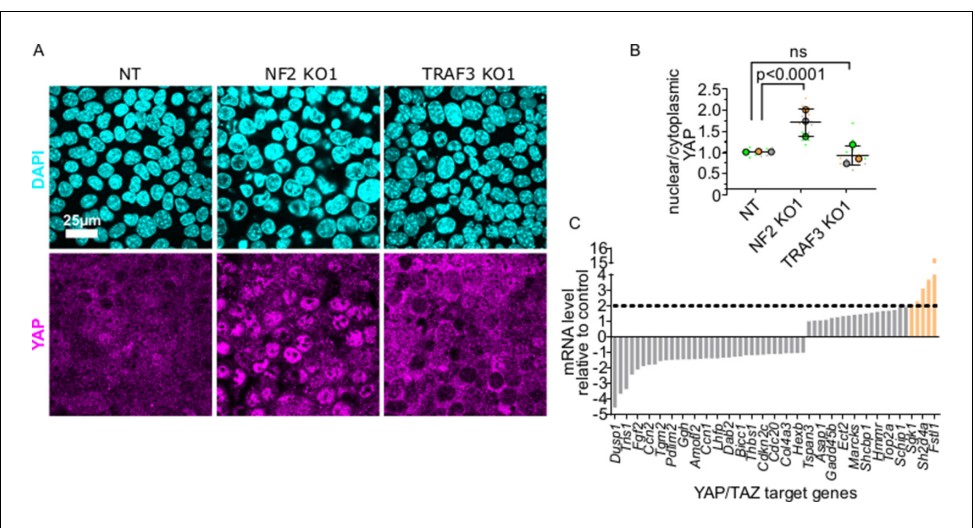

**Figure 7.** YAP/TAZ signaling is not activated by loss of TRAF3. (**A**) NT, NF2 KO1, and TRAF3 KO1 cells stained for YAP and DAPI. (**B**) Quantifications of nuclear to cytoplasmic YAP ratio in NT, NF2 KO1, and TRAF3 KO1 cells. Measurements were of at least 12 fields of view per condition per experimental replicate, and the average ratio was calculated. Data are presented as a SuperPlot (n = 3). p-values were calculated by mixed model two-way ANOVA. (**C**) Conserved YAP/TAZ gene signature expression in TRAF3 KO1 cells shown as fold differences between NT control and TRAF3 KO1. Dashed threshold line is based on 2× increase in gene expression.

The online version of this article includes the following source data and figure supplement(s) for figure 7:

**Source data 1.** Source data file for *Figure 7*.

**Figure supplement 1.** TRAF3 KO cells over-proliferate cell autonomously.

**Figure supplement 1—source data 1.** Source data file for *Figure 7—figure supplement 1*.

**Figure supplement 2.** Loss of Traf3 does not affect CKI levels.

**Figure supplement 2—source data 1.** Source data file for *Figure 7—figure supplement 2*.

*1A and B*). Therefore, we asked if the noncanonical NF-κB pathway suppresses CKI induction at high cell density. We immunoblotted for p27 and other CKIs including CDKN1A (p21), CDKN2C (p18), and CDKN2D (p19), in subconfluent and dense cultures (4 days post-confluency) of NT, NT/p100 KO, TRAF3 KO1, and TRAF3/p100 double KO cells (*Figure 7—figure supplement 2A–C*). p21 levels were low in cells at low density, but were high in all dense cultures, irrespective of TRAF3 or p100 expression. No significant changes in any of the other CKIs we tested were detected at low versus high cell density, or in the absence of TRAF3 or p100 (*Figure 7—figure supplement 2A'–C'*). These data argue that noncanonical NF-κB signaling overrides the ability of these CKIs to induce cell cycle arrest in EpH4 cells.

## Loss of TRAF3 blocks cells from entering G0

It was previously shown that TRAF3 induces over-proliferation by triggering CYCLIN D1 expression (*Demicco et al., 2005*; *Park et al., 2006*; *Rocha et al., 2003*; *Zhang et al., 2007*; *Cao et al., 2001*). We analyzed CYCLIN D1 in dense TRAF3 KO cells compared to NT cells by immunofluorescence. Indeed, we observed a small increase in CYCLIN D1 levels in TRAF3 KO cells compared to control (*Figure 8A and B*).

We hypothesized that loss of TRAF3 prevents cells from entering the G0 quiescent state. Through this mechanism loss of TRAF3 should give a proliferative advantage to cells not only at high density but also under other challenging conditions, such as starvation, that also promote entry into G0. To test this idea, we grew NT and TRAF3 KO cells in normal medium or medium devoid of FBS (*Figure 8C*). We observed no difference in proliferation between NT and TRAF3 KO cells in normal medium after 1 day post-confluency. However, starvation decreased proliferation in control cells but not in TRAF3 KO cells. These data suggest that loss of TRAF3 prevents cells from entering G0.

To further explore this possibility, we labeled control and TRAF3 KO cells with markers to differentiate quiescent and cycling cells. Cells exiting the cell cycle and entering G0 are known to have very low CDK2 activity. To determine if loss of TRAF3 prevents entry into G0, we used a previously validated biosensor, DHB-mVenus, which can quantitatively reveal CDK2 activity in live cells based on its distribution between the nucleus (G0/G1) and cytoplasm (G2/M) (*Figure 8D*; *Spencer et al., 2013*; *Gookin et al., 2017*). As shown in *Figure 8E and F*, at high density control cells enter quiescence, with almost exclusive nuclear localization of the biosensor. In contrast, cells lacking TRAF3 show an increased percent of cytoplasmic DHB-mVenus localization at high density, consistent with them continuing in cycle instead of entering quiescence.

Finally, we stained the cells for phospho-RB (Ser807/811). The retinoblastoma protein is a central regulator of the cell cycle and becomes dephosphorylated in G0, but is progressively phosphorylated and inactivated as cells leave G0 and move through G1 into S phase (*Figure 8D*; *Spencer et al., 2013*; *Gookin et al., 2017*). RB phosphorylation was very low in NT control at high density but loss of TRAF3 induced a dramatic increase in phospho-RB positive cells (*Figure 8E and G*). Together, these data strongly argue that the key effect of loss of TRAF3 is a failure to enter G0 under conditions that normally promote quiescence.

## Discussion

Epithelial cells possess intrinsic mechanisms to achieve and maintain an appropriate cell density, which is essential for normal tissue morphogenesis and maintenance (*Fomicheva et al., 2020*). Mechanisms of homeostatic cell density establishment are not fully understood. One of the most recognized pathways controlling cell density-dependent proliferation is the Hippo/YAP signaling pathway (*Dupont et al., 2011*). However, there are other mechanisms that control cell density independently from Hippo/YAP; for instance, LGL1/2 controls p27 levels in a density-dependent manner (*Yamashita et al., 2015*). With the goal of identifying novel components that control homeostatic cell density, we used an unbiased genome-wide CRISPR KO screen, in which we selected for cells that over-proliferate at high density. We uncovered a known (NF2) and two novel (TRAF3 and UBE2M) regulators of epithelial cell density. Loss of any of these genes results in increased cell proliferation at high density and multilayering of cells; however, proliferation at low cell density remains unaltered.

TRAF3 was of particular interest because this protein is essential for blood cell expansion and regulation of the immune response, but its functions in other tissues still remain unclear, although it is

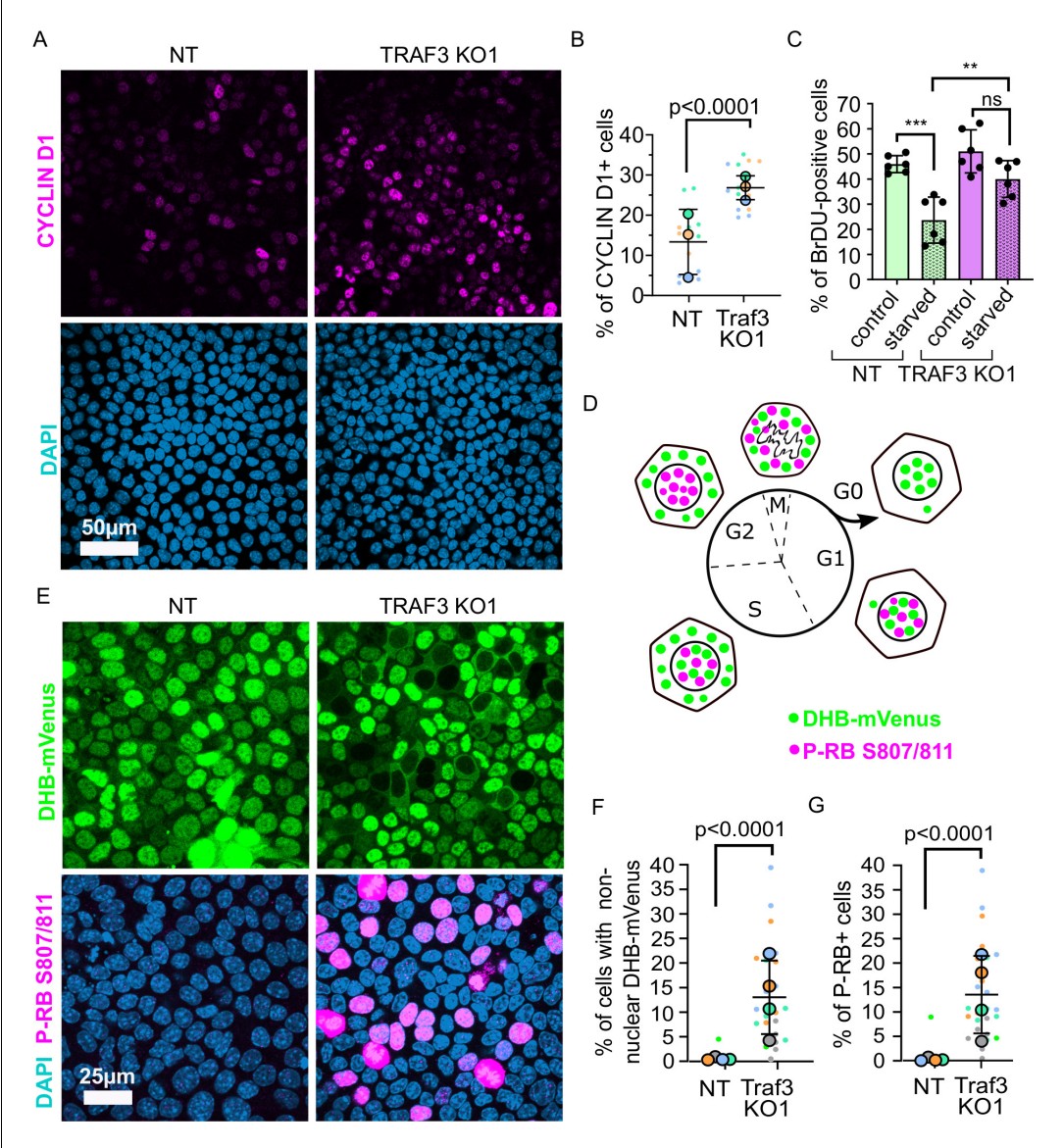

**Figure 8.** Loss of TRAF3 prevents cells from entering G0. (**A**) NT and TRAF3 KO1 cells stained for CYCLIN D1 and DAPI. (**B**) Quantifications of CYCLIN D1 positive cells in NT and TRAF3 KO cells. Data are presented as a SuperPlot (n = 3). p-value was calculated by mixed model two-way ANOVA. (**C**) NT and TRAF3 KO1 cells grown for 24 hr in regular media or media without FBS (starved) were treated with BrdU and analyzed by flow cytometry. (**D**) A schematic of DHB-mVenus and phospho-RB S807/811 localization at different phases of cell cycle. (**E**) NT and TRAF3 KO1 cells stably expressing DHB-mVenus were grown to 4 days post-confluency and stained for phospho-RB S807/811 and DAPI. (**F**) Quantifications of cells with non-nuclear DHB-mVenus localization. Data are presented as a SuperPlot (n = 4). p-value was calculated by mixed model two-way ANOVA. (**G**) Quantification of phospho-RB positive cells. Data are presented as a SuperPlot (n = 4). p-value was calculated by mixed model two-way ANOVA.

The online version of this article includes the following source data for figure 8:

**Source data 1.** Source data file for *Figure 8*.

ubiquitously expressed (*Yue et al., 2014*). We discovered that loss of TRAF3 results in cell over-growth in both mouse and human mammary gland cell lines, and mesenchymal cells (NIH 3T3 fibro-blasts) as well as in primary mouse mammary gland organoids. We interrogated the mechanism of TRAF3-dependent proliferation control in mammary gland epithelial cells (*Figure 9*) and found that loss of TRAF3 specifically activates the noncanonical branch of NF-κB signaling, while having no effect on canonical NF-κB signaling. We further discovered that the noncanonical pathway is necessary for over-proliferation at high density in cells lacking TRAF3.

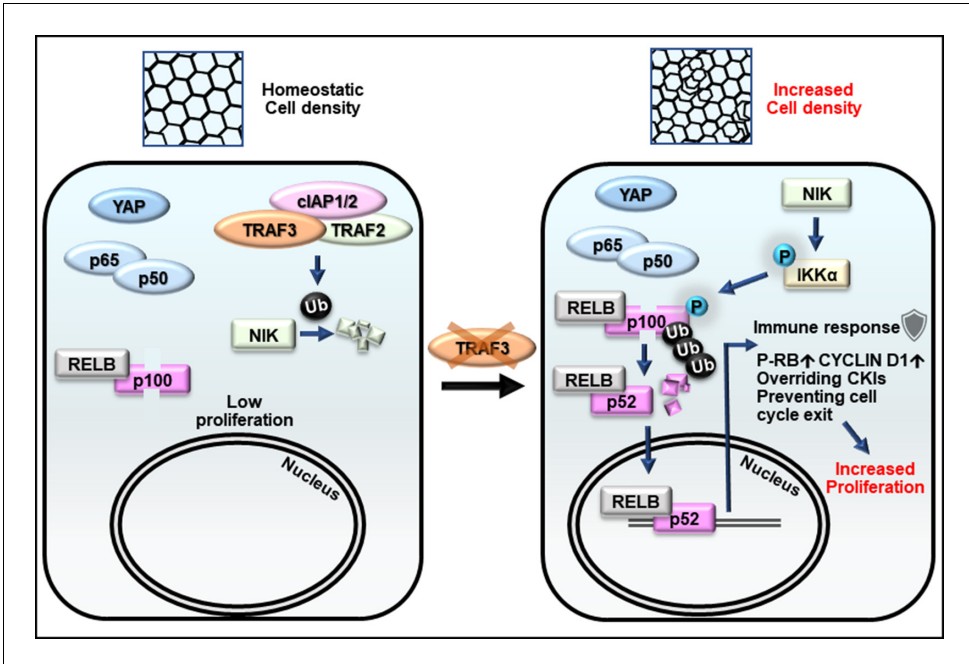

**Figure 9.** Model for the role of TRAF3 in density-dependent proliferation. Cells at high density rarely proliferate (on the left). They have active Hippo signaling (YAP is cytoplasmic), inactive canonical NF-κB (p65/p50 is cytoplasmic) and inactive noncanonical NF-κB signaling (NIK is low, p100 is not processed to p52, and RELB is cytoplasmic). Upon loss of TRAF3, Hippo and canonical NF-κB signaling remain unaltered, but noncanonical NF-κB signaling is activated (on the right). NIK levels rise in the absence of TRAF3, activates the downstream cascade, resulting in increased p100 processing to p52. The RELB/p52 complex activates innate immune response genes, including antigen presentation and interferon pathway components. It also overrides CKIs, induces CYCLIN D1 expression, and prevents entering G0 as shown by increased number of cells with active CDK2 and phosphorylated RB. Altogether it results in cell over-proliferation at high density.

Our RNAseq data revealed that loss of TRAF3 does not induce the transcription of classical proliferation pathway components, including YAP/TAZ, PI3K/AKT, MAPK, TGF-β, and WNT signaling. We also demonstrated that over-proliferation caused by loss of TRAF3 is cell autonomous, as control cells did not over-proliferate when co-cultured with TRAF3 KO cells. Together, these data suggest that noncanonical NF-κB signaling does not positively drive proliferation but instead suppresses the ability of cells to exit the cell cycle. One key exit mechanism operates through induction of CKIs. For instance, in some epithelial cells p27 levels increase with density and high levels of p27 inhibit cell proliferation (*Yamashita et al., 2015*). However, we found no differences in levels of p27 or other CKIs (p21, p19, and p18) in cells lacking TRAF3 versus control EpH4 cells, although p21 levels do increase with increased cell density. Therefore, either EpH4 cells do not respond to CKIs or constitutive activation of the noncanonical NF-κB pathway overrides CKI-mediated cell cycle exit. Nonetheless, our data demonstrate that loss of TRAF3 prevents cells from entering quiescent state G0 as shown by increased CDK2 activity and RB phosphorylation.

RNAseq analysis revealed that loss of TRAF3 leads to the activation of immune response pathways, at least in part through noncanonical NF-κB signaling. Activated genes include antigen presentation, interferon response pathways, and genes responsible for foreign DNA and RNA recognition. Expression of these genes was reduced when p100 was deleted in the TRAF3 KO cells. However, certain genes induced in the TRAF3 KO cells, including adhesion molecules, superoxide-producing enzymes, and chemokines, were not reduced by deletion of p100. These data reveal that a mechanism independent of noncanonical NF-κB signaling can drive gene expression responses activated by loss of TRAF3.

The literature suggests that the loss or reduction of *Traf3* expression or activation of noncanonical NF-κB signaling might be important for hyperplasia in response to infection or contribute to cancer

development. It is known that epithelial cells can respond to infection by mounting an innate immune response. The noncanonical NF-κB pathway can be activated by contact with pathogens, and bronchial, intestinal, and mammary gland epithelia can each express a variety of innate immune factors that protect the organisms against infectious agents and attract immune cells (*Strandberg et al., 2005*; *Philpott et al., 2001*). *Helicobacter pylori*, a bacterium associated with the development of gastritis and gastric cancer, activates noncanonical NF-κB signaling in a stomach epithelial cell line and in *H. pylori*-associated gastritis (*Feige et al., 2018*). Moreover, activation of the noncanonical NF-κB pathway in response to infection correlates with hyperplasia of the ear mucosa following otitis infection, although the mechanism was not investigated (*Cho et al., 2016*).

As discussed above, *Traf3* is mutated, though at low frequency, in different cancers of epithelial origin (*Zhu et al., 2018*), and low expression is associated with worse outcomes. Noncanonical NF-κB signaling is also associated with different epithelial cancers, including breast cancer (*Sovak et al., 1997*; *Rojo et al., 2016*) and lung cancer (*Dimitrakopoulos et al., 2019*). Further work will be required to determine the impact of this pathway in cancer development.

# Materials and methods

## Key resources table

| Reagent type (species) or resource | Designation | Source or reference | Identifiers | Additional information |
|---|---|---|---|---|
| Recombinant DNA reagent (*Mus musculus*) | GeCKO v2 KO Pooled Library | *Sanjana et al., 2014*, PMID:25075903; Addgene | CAT: 1000000052 | |
| Recombinant DNA reagent (*Mus musculus*) | Scrambled control shRNA | Sigma MISSION shRNA library | CAT: SHC016-1EA | |
| Recombinant DNA reagent (*Mus musculus*) | The p27 shRNA | Sigma MISSION shRNA library | CAT: TRCN0000287390 | |
| Recombinant DNA reagent (*Mus musculus*) | p27-qPCR-FWD primer | This paper | N/A | TCAAACGTGAGAGTGTCTAACG |
| Recombinant DNA reagent (*Mus musculus*) | p27-qPCR-REV primer | This paper | N/A | CCGGGCCGAAGAGATTTCTG |
| Recombinant DNA reagent (*Mus musculus*) | Puromycin-qPCR-FWD primer | This paper | N/A | CTGCAAGAACTCTTCCTCACG |
| Recombinant DNA reagent (*Mus musculus*) | Puromycin-qPCR-REV primer | This paper | N/A | GGGAACCGCTCAACTCGG |
| Recombinant DNA reagent (*Mus musculus*) | Control non-targeting (NT) sgRNA | This paper | N/A | GCGAGGTATTCGGCTCCGCG |
| Recombinant DNA reagent (*Mus musculus*) | Nf2 sgRNA KO1 | This paper | N/A | CGAGATGGAGTTCAACTGCG |
| Recombinant DNA reagent (*Mus musculus*) | Nf2 sgRNA KO2 | This paper | N/A | ATACTGCAGTCCAAAGAACC |
| Recombinant DNA reagent (*Mus musculus*) | Traf3 sgRNA KO1 | This paper | N/A | GTGCTCGTGCCGGAGCAAGG |
| Recombinant DNA reagent (*Mus musculus*) | Traf3 sgRNA KO2 | This paper | N/A | TGGCCCTTCAGGTCTACTGT |

*Continued on next page*

*Continued*

| Reagent type (species) or resource | Designation | Source or reference | Identifiers | Additional information |
|---|---|---|---|---|
| Recombinant DNA reagent (*Mus musculus*) | Ube2m sgRNA KO1 | This paper | N/A | GCGCAGCTCC GGATTCAGAA |
| Recombinant DNA reagent (*Mus musculus*) | Ube2m sgRNA KO2 | This paper | N/A | GAGTCGGCCGG CGGCACCAA |
| Recombinant DNA reagent (*Mus musculus*) | Nfkb2/p100 sgRNA KO1 | This paper | N/A | CTGAGCGTGA TAAATGACGT |
| Recombinant DNA reagent (*Mus musculus*) | Nfkb2/p100 sgRNA KO2 | This paper | N/A | CTGTTCCACAA TCACCAGAT |
| Recombinant DNA reagent (*Mus musculus*) | Map3k14/NIK sgRNA KO1 | This paper | N/A | TCAGAGCGCA TTTTCATCGC |
| Recombinant DNA reagent (*Mus musculus*) | Map3k14/NIK sgRNA KO2 | This paper | N/A | GTCGAGGCAG TACCGGTCGC |
| Recombinant DNA reagent (*Homo sapiens*) | Traf3 sgRNA KO1 | This paper | N/A | AGATTCGCGA CTACAAGCGG |
| Recombinant DNA reagent (*Homo sapiens*) | Traf3 sgRNA KO2 | This paper | N/A | CCTCACATGTT TGCTCTCGC |
| Recombinant DNA reagent (*Homo sapiens*) | DHB-mVenus | *Spencer et al., 2013*, PMID:24075009; Addgene | CAT: 136461, RRID:Addgene_136461 | |
| Recombinant DNA reagent (synthetic) | ES-FUCCI | *Sladitschek and Neveu, 2015*, PMID:25909630; Addgene | CAT: 62451, RRID:Addgene_62451 | |
| Recombinant DNA reagent (*Mus musculus*) | pWPI mScarlet NIK-ΔT3 | This paper | N/A | Mouse NIK with deleted TRAF3 binding motif (amino acids 78–84) cloned into pWPI-mScarlet vector |
| Antibody | Anti-NF2/MERLIN (rabbit monoclonal) | CST | CAT: 6995S, RRID:AB_10828709 | (1:750), WB |
| Antibody | Anti-TRAF3 (mouse monoclonal) | Santa Cruz | CAT: sc-6933, RRID:AB_628390 | (1:200), WB |
| Antibody | Anti-UBE2M (rabbit polyclonal) | Proteintech | CAT: 14520–1-AP | (1:500), WB |
| Antibody | Anti-NF-κB2 p100/p52 (rabbit polyclonal) | CST | CAT: 4882, RRID:AB_10695537 | (1:750), WB |
| Antibody | Anti-NIK (rabbit polyclonal) | CST | CAT: 4994S, RRID:AB_2297422 | (1:500), WB |
| Antibody | Anti-RELB (rabbit monoclonal) | Abcam | CAT: ab180127 | (1:1000), WB; (1:300), IF |
| Antibody | Anti-NF-kappaB Pathway Sampler Kit Antibody | CST | CAT: 9936, RRID:AB_561197 | (1:1000), WB |
| Antibody | Anti-CDKN1B/ p27 (rabbit polyclonal) | BD Transduction Laboratories | CAT: 610241, RRID:AB_610241 | (1:1000), WB |

*Continued*

| Reagent type (species) or resource | Designation | Source or reference | Identifiers | Additional information |
|---|---|---|---|---|
| Antibody | Anti-p21 (mouse monoclonal) | Invitrogen | CAT: ma5-14353, RRID:AB_10986834 | (1:1000), WB |
| Antibody | Anti-p19 (rabbit polyclonal) | Santa Cruz | CAT: sc-1063, RRID:AB_2078865 | (1:200), WB |
| Antibody | Anti-p18 (rabbit polyclonal) | Santa Cruz | CAT: sc-1064, RRID:AB_2078729 | (1:200), WB |
| Antibody | Anti-GAPDH (rabbit monoclonal) | CST | CAT: 2118S, RRID:AB_561053 | (1:2000), WB |
| Antibody | Anti-α-tubulin (mouse monoclonal) | Sigma-Aldrich | CAT: T-9026, RRID:AB_477593 | (1:4000), WB |
| Antibody | Anti-actin (mouse monoclonal) | Sigma-Aldrich | CAT: A4700, RRID:AB_476730 | (1:4000), WB |
| Antibody | Anti-BrdU antibodies (rat monoclonal) | Abcam | CAT: ab6326, RRID:AB_305426 | (1:800), IF, Flow |
| Antibody | Anti-YAP (rabbit polyclonal) | Novus Biologicals | CAT: NB110-58358, RRID:AB_922796 | (1:200), IF |
| Antibody | Anti-phospho-RB Ser807/811 (rabbit monoclonal) | CST | CAT: 8516S, RRID:AB_11178658 | (1:1000), IF |
| Antibody | Anti-CYCLIN D1 (mouse monoclonal) | Invitrogen | CAT: MA5-11387, RRID:AB_10987096 | (1:50), IF |
| Antibody | Anti-KI-67 (rabbit polyclonal) | Invitrogen | CAT: 18-0191Z, RRID:AB_86661 | 1:70, IF |
| Antibody | Anti-Phospho-HISTONE H3 (Ser10) (mouse monoclonal) | CST | CAT: 9706S, RRID:AB_331748 | 1:300, IF |
| Antibody | Anti-E-CADHERIN (rat monoclonal) | Thermo Fisher Scientific | CAT: 14-3249-80, RRID:AB_1210459 | 1:500, IF |
| Others | Hoechst 33342 Fluorescent dye | Life Technologies | CAT: 62249 | 1:1000, IF |
| Others | DAPI Fluorescent dye | Sigma-Aldrich | CAT: 422801 | 1:500, IF |
| Others | DRAQ5 Fluorescent dye | CST | CAT: 4084 | 1:1000, IF |
| Cell line (*Mus musculus*) | EpH4 | Dr. Jurgen Knoblich, Institute of Molecular Biotechnology, Vienna, Austria | | Identity verified by RNAseq, DNA sequencing and immunofluorescent staining for epithelial markers |
| Cell line (*Homo sapiens*) | HEK293T | ATCC | RRID:CVCL_0063 | |
| Cell line (*Homo sapiens*) | MCF10a | ATCC | RRID:CVCL_0598 | |
| Cell line (*Mus musculus*) | NIH 3T3 | ATCC | RRID:CVCL_0594 | |

*Continued on next page*

*Continued*

| Reagent type (species) or resource | Designation | Source or reference | Identifiers | Additional information |
|---|---|---|---|---|
| Strain (*Mus musculus*) | C3H/HeNCrl | Charles River Laboratories | CAT: CRL:025, RRID:IMSR_CRL:025 | |
| Chemical compound, drug | BV6 | ApexBio | CAT: B4653 | |
| Software, algorithm | GraphPad Prism | GraphPad | RRID:SCR_002798 | |
| Software, algorithm | CLC Genomics Workbench | QIAGEN | RRID:SCR_011853 | |
| Software, algorithm | Ingenuity Pathway Analysis (Qiagen) | QIAGEN | SCR_008653 | |
| Software, algorithm | Fiji is just ImageJ | N/A | RRID:SCR_002285 | |
| Software, algorithm | MAGeCK software | *Li et al., 2014*, PMID:25476604 | N/A | |

## Whole-genome CRISPR KO screen

Mouse CRISPR GeCKO v2 Knockout Pooled Library (*Shalem et al., 2014*; *Sanjana et al., 2014*) was purchased from Addgene. The library was amplified according to the developer's protocol (*Shalem et al., 2014*; *Sanjana et al., 2014*). Library lentiviruses were produced as described below. EpH4-FUCCI cells transduced with the library were grown for 10 days to allow time for gene-editing and depletion of the target proteins. The cells were then seeded at 100,000 cells/cm$^2$ and cultured for 4 days, then trypsinized with 0.25% trypsin (Gibco) and $4 \times 10^7$ cells (about 300 cells per sgRNA) were sorted on a FACSAria III for mCitrine+ (proliferating) cells. This population was replated, expanded, and resorted for mCitrine+ cells. This process was repeated for a total of three sorts. Genomic DNA from $4 \times 10^7$ cells before sorting, and after the first, second, and third rounds of sorting was purified using Blood and Cell Culture DNA Midi Kit (Qiagen). The sgRNA sequences were PCR amplified ($26 \times 100$ µl reactions) from genomic DNA (~260 µg) using adaptor primers developed by the Zhang laboratory. The products of the first PCR reactions were amplified again with Illumina index primers to add barcodes and Illumina adaptors. The products of this reaction were purified and sequenced on a Novaseq instrument. We utilized MAGeCK software (*Li et al., 2014*) to analyze the fastq files. A read count distribution graph was generated using RStudio.

## Plasmid constructs and primers

ES-FUCCI (plasmid # 62451; *Sladitschek and Neveu, 2015*) and DHB-mVenus (plasmid #136461; *Spencer et al., 2013*) were purchased from Addgene. The *p27* shRNA clone TRCN0000287390 in pLKO1 was obtained from the Sigma MISSION shRNA library. *p27* knockdown was tested by qPCR and compared to cells transduced with a scrambled non-targeting control shRNA (*Figure 1—source data 1*).

The sgRNAs used in this study are listed in the Key Resources Table. sgRNAs were cloned into lentiCRISPR v2 vector at the *BsmB*I restriction site using Zhang lab protocol (*Shalem et al., 2014*; *Sanjana et al., 2014*). p100 sgRNAs were cloned into a modified pLVTHM GFP vector between *Cla*I and *Mlu*I sites or into lentiCRISPR v2 vector.

Deletion of TRAF3 binding motif (amino acids 78–84) from wild-type NIK cDNA was performed by site-directed mutagenesis. NIK-ΔT3 cDNA was cloned into the pWPI mScarlet vector between *BamH*I and *Asc*I sites.

## Cell culture, lentiviral transductions, transfections, and chemicals

EpH4 cells were obtained from Dr. Jurgen Knoblich (Institute of Molecular Biotechnology, Vienna, Austria). HEK293T, MCF10a, and NIH 3T3 were obtained from the ATCC. Cells are routinely tested for mycoplasma contamination by Hoechst DNA staining, and all were found to be negative. EpH4,

NIH 3T3, and HEK293T cells were cultured in DMEM (Gibco) supplemented with 10% fetal bovine serum (Atlanta Biologicals) and 1× penicillin/streptomycin (Life Technologies). MCF10a cells were cultured in DMEM/F12 media (Gibco) supplemented with 5% horse serum (Gibco), EGF 20 ng/ml (Sigma), hydrocortisone 0.5 mg/ml (Sigma), cholera toxin 100 ng/ml (MP Biomedicals), insulin 10 µg/ml (Sigma), and 1× penicillin/streptomycin (Life Technologies). EpH4 cells were seeded at 100,000 cells/cm² and cultured for 1 or 4 days. MCF10a cells were seeded at 100,000 cells/cm² and cultured for 5 days. Lentiviruses were produced by calcium phosphate transfection of HEK293T cells with lentivectors and lentiviral packaging vectors psPAX2 and pMD2.G. Lentiviruses were collected and concentrated 48 hr after transduction, utilizing Amicon centrifugal filter units. EpH4 and MCF10a cells were transduced by lentiviruses by shaking cells and viruses in suspension at 400 rpm, 37°C for 1 hr. Transduced cells were selected by puromycin (CRISPR v2 vectors) or FACS (pWPI mScarlet, pLVTHM GFP, and DHB-mVenus vectors). EpH4 cells used for mixing experiments were NT pWPI mScarlet, NT pLVTHM GFP, and *Traf3* KO1 pWPI mScarlet. SMAC mimetic BV6 was purchased from ApexBio (B4653).

To develop the EpH4 ES-FUCCI stable cell line we linearized the ES-FUCCI vector with *AseI* and transduced cells using Xfect (Clontech) according to the manufacturer's protocol. Cells were selected with 300 µg/ml Hygromycin followed by FAC sorting of mCitrine+ and/or mCherry+ cells.

## Flow cytometry analysis

Cells were treated with 3 µg/ml BrdU (Sigma-Aldrich) for 1 hr, then washed in sterile PBS and trypsinized. Trypsin was blocked with complete medium plus 50 µg/ml DNase to reduce clumping of cells. Cells were pelleted, incubated in 5 mM EDTA in PBS on ice for 10 min, and fixed in 4% paraformaldehyde (PFA) for 15 min at room temperature. BrdU incorporation was detected after treatment with 2N HCl for 20 min at 37°C, washing with 1.5 M Na2B4O7, blocking in Western Blocking reagent (Roche), and staining with anti-BrdU antibodies (1:800, Abcam, ab6326). Cells were analyzed using a Fortessa flow cytometer and analyzed using FlowJo.

## Immunoblotting

Cells were lysed in buffer containing 0.1% Triton-X100, 20 mM HEPES (pH 7.4), 50 mM NaCl, and 2 mM EDTA supplemented with a protease inhibitor cocktail (Roche), Calyculin A, and PhosStop phosphatase inhibitors (Roche). Cell lysates were briefly sonicated and centrifuged at 16,100 g for 10 min. After centrifugation, the soluble fraction was boiled with SDS sample buffer for 5 min. Antibodies used for western blotting are listed in the Key Resources Table.

## Immunofluorescence staining, image acquisition, and analysis

Cells were grown on LabTek II chamber slides (Thermo Scientific) for the indicated times and fixed with 4% PFA at room temperature for 15 min. Cells were permeabilized with 0.2% Triton X-100, blocked in 1× Western Blocking Reagent (Roche), and labeled for IF imaging. Primary antibodies used for IF are listed in the Key Resources Table. Secondary labeling was performed using Alexa Fluor secondary antibodies (1:500-1:1000, Life Technologies). Samples were mounted using Fluoromount G (Electron Microscopy Sciences). Laser scanning confocal images were acquired using 20×/0.75 Plan Apo or 100×/1.40 Plan Apo oil immersion objectives on a Nikon A1R inverted confocal microscope (Nikon Instruments Inc). Epifluorescence images were acquired using an EVOS FL inverted microscope (Life Technologies). Fiji software was used for post-acquisition processing.

Fiji Temporal Color Code function (fire color scale) was used for depth color coding of cells for visualization of multilayering. To measure the percent of multilayering in KO cells, we used Fiji software to process confocal image stacks of cells that had been grown to high density and stained for a nuclear marker. A single slice above the monolayer (about 8 µm above the center of the nuclei in the first layer of cells) was processed using Gaussian blur and Threshold to create a binary mask. It was analyzed using Measure particles function to determine the percent of total area that is above the monolayer. Data were collected for five fields of view in three biological repeats.

Nuclear/cytoplasmic ratio of YAP was measured using the ImageJ Intensity Ratio Nuclei CytoplasmTool (RRID:SCR_018573; https://github.com/MontpellierRessourcesImagerie/imagej_macros_and_scripts/wiki/Intensity-Ratio-Nuclei-Cytoplasm-Tool).

## Real-time qPCR

Total RNA was extracted with TRIzol (Life Technologies). cDNA was reverse transcribed using the SuperScript III First-Strand Synthesis System (Invitrogen). qPCR was performed with triplicate replicates on a BioRad CFX96 Thermocycler and analyzed using the ΔΔCt method. Expression levels were calculated relative to *Gapdh*. *p27* levels were assessed using *p27* primers. The primers are listed in the Key Resources Table. The enrichment of cells containing sh-*p27* after FACS was assessed using puromycin resistance gene primers, because the Puro cassette was integrated into the cell genome together with the *p27* shRNA.

## RNA sequencing

NT, TRAF3 KO1, and TRAF3/p100 double KO EpH4-FUCCI cells were sorted for non-proliferating cells (mCherry+) by FACS. RNA from NT, TRAF3 KO1, and TRAF3/p100 double KO cells from two experimental repeats was isolated using the RNeasy Mini Kit (Qiagen). RNA quality control was performed using a 2100 Bioanalyzer (Agilent Technologies). Samples had RNA integrity numbers in the range 8.1–9.3. Sequencing was performed using the NovaSeq 6000 Sequencing System (Illumina, San Diego, CA). Data were processed and analyzed with CLC Genomics Workbench and Ingenuity Pathway Analysis (Qiagen).

## Mouse mammary gland organoids

Mammary glands were isolated from 8-week-old C3H mice as described previously (*Pasic et al., 2011*). Cells were then briefly treated with 0.25% trypsin and filtered through a 40 μm strainer. Primary mammary cells were transduced with lentiCRISPR v2 NT control and *Traf3* KO1 and grown for 8 days in 50% Matrigel (Corning) supplemented with Organoid Growth Medium (DMEM/F12 (Gibco), 5 ng/ml EGF (Sigma), 3 ng/ml mFGF2 (R and D Systems), and 1× ITS (Millipore Sigma)). Then organoids were fixed with PFA and immunostained. All mouse experimental procedures were approved by Vanderbilt Institutional Animal Care and Use Committee; IACUC protocol number M1800045, Exp: 04/26/2021.

## Statistical analysis

Data were tested for normality by Shapiro–Wilk test and then analyzed by Student's t-test, one-way ANOVA, or mixed model two-way ANOVA statistical test using GraphPad Prism software. When using ANOVA, post hoc analysis was done using Tukey or Dunnett multiple comparison tests. All statistical analyses were considered significant at $p<0.05$. Data presented as SuperPlots (*Lord et al., 2020*) combine individual measurements (small dots color coded for each replicate), mean values for each replicate (large dots color coded for each replicate), and mean ± s.d.

## Acknowledgements

This work was supported by NCI grant R35CA132898 to IGM. We would like to thank the members of the Macara lab for their valuable feedback, as well as Christian de Caestecker, Christian Meyer, and Daria Episheva for the help with processing Illumina data for the CRISPR screen. We also thank Vivian Gama and Edward Levine for sharing antibodies. Flow cytometry experiments were performed in the VMC Flow Cytometry Shared Resource; Illumina sequencing and RNAseq were performed by Vanderbilt Technologies for Advanced Genomics. These resources are supported by the Vanderbilt Ingram Cancer Center (P30 CA68485) and the Vanderbilt Digestive Disease Research Center (DK058404).

## Additional information

### Funding

| Funder | Grant reference number | Author |
| --- | --- | --- |
| National Cancer Institute | R35 CA132898 | Ian G Macara |

The funders had no role in study design, data collection and interpretation, or the decision to submit the work for publication.

### Author contributions
Maria Fomicheva, Data curation, Formal analysis, Validation, Investigation, Visualization, Methodology, Writing - original draft, Writing - review and editing; Ian G Macara, Conceptualization, Resources, Supervision, Funding acquisition, Methodology, Project administration, Writing - review and editing

### Author ORCIDs
Maria Fomicheva (iD) https://orcid.org/0000-0002-0281-0467
Ian G Macara (iD) https://orcid.org/0000-0001-8546-5357

### Ethics
Animal experimentation: All mouse experimental procedures were approved by Vanderbilt Institutional Animal Care and Use Committee; IACUC protocol number M1800045, Exp: 04/26/2021.

### Decision letter and Author response
Decision letter https://doi.org/10.7554/eLife.63603.sa1
Author response https://doi.org/10.7554/eLife.63603.sa2

## Additional files

### Supplementary files
• Transparent reporting form

### Data availability
Sequencing data have been deposited in GEO under accession code GSE147767 All other data generated or analysed during this study are included in the manuscript and supporting files.

The following dataset was generated:

| Author(s) | Year | Dataset title | Dataset URL | Database and Identifier |
|---|---|---|---|---|
| Fomicheva M, Macara IG | 2020 | RNA sequencing of NT control cells, Traf3 KO, and Traf3/p100 double KO cells | https://www.ncbi.nlm.nih.gov/geo/query/acc.cgi?acc=GSE147767 | NCBI Gene Expression Omnibus, GSE147767 |

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
