## [Decision Letter]

[Editors' note: this paper was reviewed by Review Commons.]

**Acceptance summary:**

This study reports the identification of Traf3, a negative regulator of NF-κB signaling, as a novel regulator of density-dependent proliferation from a cell-based CRISPR screen. The authors further show that loss of Traf3 specifically activates non-canonical NF-κB signaling, which in turn triggers an innate immune response and drives cell division independently of known density-dependent proliferation mechanisms, including YAP/TAZ signaling and cyclin kinase inhibitors. Overall, this is a well-executed study that has revealed novel insights into the control of epithelial cell proliferation.

---

## [Author Response]

Reviewer #1This is an interesting manuscript which is utilizing comprehensive CRISPR lossof-function screen in conjunction with cell cycle fluorescence reporter (FUCCI) to identify novel regulators underlying cell density dependent proliferation of mouse/human epithelium cells. Besides know regulators, Traf3, a negative regulator of NF-κB signaling was revealed to functionally involve in density-dependent proliferation regulation in both mouse and human epithelium cell lines and primary cell derived organelles. They further conducted validation, signaling pathway analysis and transcriptome characterization. And they proposed that loss of Traf3 specifically activates non-canonical NF-κB signaling, which in turn triggers an innate immune response. The Traf3 seems to work independent of canonical YAP/TAZ signaling and cyclin kinase inhibitors. Overall, this is a well-designed and appropriately executed study. The data were collected with proper controls and largely supports the conclusions drawn. Several key issues should be clarified to further strengthen the manuscript as follows:Major points1) It has been known that unlike other Traf family members, Traf3 plays an essential role in regulating the noncanonical NF-κB activity in vivo and in vivo, which has been studies in other systems (such as J Biol Chem. 2007 Feb 9;282(6):3688-94). That's being said, mechanism novelty is less empowered.

We were aware that the function of TRAF3 as a negative regulator of non-canonical NFκB signaling is well known, but the novelty of our study concerns the impact of this pathway on density-dependent cell proliferation, which we believe to be novel and unexpected.

2) The largest part of manuscript heavily relies on the EpH4 cell line including the CRISPR screen, validation and mechanism study. Although human MCF10A and primary mouse organelles were used to confirm the phenotype, it is still not quite clear whether the phenotype presented here is specific to epithelial cells. More cell lines and control cell lines should be included to clarify this point.

We agree, and in response we have knocked out *Traf3* in NIH 3T3 fibroblasts, as an example of a non-epithelial cell line. We observed that TRAF3 KO 3T3 cells lose contact inhibition and over-proliferate in dense monolayers, compared to control (Figure 3—figure supplement 1C-D’).

3) It's interesting such screening did not identify typical tumor suppressors such as Trp53. I then realize that EpH4 is an immortalized mouse mammary gland epithelial cell line (Fialka et al., 1996), which may undergo selective deletion or mutation of Trp53 or Rb1. If so, the dispensable role of cyclin kinase inhibitors is expected. Authors should characterize the cell line. Again, as stated in 2#, more cell lines and controls should be validated.

Cell cycle arrest at high density is not normally controlled by Trp53, so it is unsurprising that this gene did not show up in our screen. Although the EpH4 cell might not contain functional p53, we confirmed the over-proliferation phenotype and activation of noncanonical NF-κB in primary mammary gland organoids. This result demonstrates that the response is not dependent on mutant p53.

4) I will not recommend the tone of "Epithelial Homeostasis" given the entire study is based on pure cell culture work unless in vivo data could be provided.

As suggested, we have changed the title to avoid the term “epithelial homeostasis”. However, we discuss homeostatic cell density control in the text of the manuscript because this term is not specific to in vivo situations but describes a collective cell behavior that is also observed in vitro.

Minor comments:1) It will be helpful to include a schematic diagram of FUCCI vector and working model in Figure 1A to keep readers who do not have related background to follow the design.

We agree and have included a schematic of FUCCI in Figure 1.

2) P value should be provided between groups in Figure 1B.

We did not provide a p value because this was a single representative experiment with 3 technical repeats, just used as a proof-of-principle experiment. We have moved these data to supplementary information section.

3) I find out redundant description of CRISPR screen and candidates, which would benefit from trimming most of statement and only focusing on Traf3 to connect to the function in the following study

We have modified the text to remove redundant descriptions.

Reviewer #1 (Significance (Required)):This is an interesting manuscript which is utilizing comprehensive CRISPR loss-offunction screen in conjunction with cell cycle fluorescence reporter (FUCCI) to identify novel regulators underlying cell density dependent proliferation of mouse/human epithelium cells. Besides know regulators, Traf3, a negative regulator of NF-κB signaling was revealed to functionally involve in density-dependent proliferation regulation in both mouse and human epithelium cell lines and primary cell derived organelles. They further conducted validation, signaling pathway analysis and transcriptome characterization. And they proposed that loss of Traf3 specifically activates non-canonical NF-κB signaling, which in turn triggers an innate immune response. The Traf3 seems to work independent of canonical YAP/TAZ signaling and cyclin kinase inhibitors. Overall, this is a well-designed and appropriately executed study. The data were collected with proper controls and largely supports the conclusions drawn. It will be of interest to the general audience.Reviewers cross commentingI agree with reviewer 2# that the current cell model barely provides anywhere any window to enhance proliferation of fast dividing cells. The authors either choose to employ stress model, or independent cells with optimal test window. Point 3# targeted on the same mechanism concern, which would benefit from additional experiment underlying whether cIAP inhibition by a SMAC-mimetic exert the same effect as TRAF3 KOReviewer #2 (Evidence, reproducibility and clarity (Required)):Fomicheva and Macara designed a CRISPR/Cas9 screen that allows the identification of genes functioning as negative regulators of density-controlled cell cycle in epithelial cells. They found that loss of TRAF3 prevents epithelial cells from halting division at high density. By increasing expression of NIK, TRAF3 ablation induces p100/p52 processing and non-canonical NF-κB signaling in epithelial cells. By generating TRAF3/NIK or TRAF3/p100 DKO cells, they demonstrate that non-canonical NF-κB activation is mediating loss of cell density control upon ablation of TRAF3. The authors present data that exclude autocrine/paracrine activation, involvement of the Hippo pathway or deregulation of cyclin-dependent kinase inhibitors (CKI) as mechanisms for loss of density control in TRAF3 KO epithelial cells.The paper describes a very interesting strategy for identifying negative regulators of cell density control and it provides a new link to non-canonical NF-κB. Technically, the manuscript is very good and the DKOs make a strong link between uncontrolled proliferation upon TRAF3 loss and non-canonical NF-κB activation. However, the functional role of TRAF3 and activation of non-canonical NF-κB remains unresolved. Further, I think the data are not convincingly showing that TRAF3 is selectively affecting epithelial homeostasis by loss or cell density-dependent cell cycle arrest. Thus, the authors need to address the following questions and concerns:Major points1) The authors suggest that the TRAF3-NIK-IKKa-p100/p52 axis is selectively regulating density-controlled cell proliferation, but I think the data are not strong enough to support this conclusion. The only data set that supports this conclusion is presented in Figure 2D – F, showing that Brdu incorporation in rapidly cycling low density cells is not enhanced by TRAF3 KO. However, while 50% of the cells are Brdu-positive at low density, only 5% are Brdu-positive at high density. Thus, there is hardly any window to enhance proliferation of fast dividing cells and TRAF3 could simply limit proliferation under restrictive conditions. Thus, the authors should determine if loss of TRAF3 enhances proliferation after serum reduction/withdrawal or other conditions of nutrient restriction. Further, it needs to be addressed if p100 and NIK loss is affecting rapidly dividing cells at low density (low or high serum). Cell cycle analyses by FACS should be performed to clearly determine at what stage of the cell cycle TRAF3 and non-canonical NF-κB are blocking/promoting cell cycle progression.

We agree and have performed serum withdrawal experiments. Serum starvation causes significant reduction in proliferation in NT control cells but not in TRAF3 KO cells (Figure 8C).

Regarding the suggestion that flow cytometry be used to determine what stage of the cell cycle is impacted, we already know from the FUCCI data that at high density the cells are predominantly in G0/G1. When Traf3 is knocked out, some of these cells exit quiescence and enter the cell cycle. We are unclear what additional information flow cytometry would provide. However, we have now tested whether the control cells are actually in G0 and if loss of Traf3 blocks entry into G0. We demonstrated this to be the case through use of a live cell CDK2 biosensor, DHB-mVenus, and by staining with antiphospho-Rb (Ser807/811) (new Figure 8). These data are explained in more detail below

2) The authors state that “… TRAF3 roles in epithelial cells has not been widely investigated….” This may be true, but there is quite an extensive literature on the role of non-canonical NF-κB in epithelial cells and also its role in cell cycle progression through the induction of cyclin D1 during G1-S phase transition (see Demicco et al., 2005, Park et al., 2006, Rocha et al., 2003, Zhang et al., 2007, Cao et al., 2001, and PMID16713561, 11713278, 20420878). My best explanation would be that p52, potentially in concert with BCL3, is inducing D-type cyclins and enforcing G1/S phase transition. This needs to be tested and it needs to be investigated if this regulation is cell density-dependent (see point 1).

This is an interesting point. We tested if TRAF3 KO cells have changes in CYCLIN D1 levels by immunofluorescent staining. Quantifications of CYCLIN D1 intensity showed modest but significant increase of CYCLIN D1 in TRAF3 KO cells (Figure 8A, B). As mentioned above, we also tested whether the loss of TRAF3 prevents cells at high density from entering G0. Cells exiting the cell cycle and entering G0 are known to have very low CDK2 activity, and to determine if loss of TRAF3 prevents entry into G0, we used a previously validated biosensor, DHB-mVenus, which can quantitatively reveal CDK2 activity in live cells based on its distribution between the nucleus (G0/G1) and cytoplasm (G2/M) (new Figure 8D). As shown in new Figure 8E and G, at high density control cells enter quiescence, with almost exclusive nuclear localization of the biosensor. In contrast, cells lacking TRAF3 show an increased percent of cytoplasmic DHB-mVenus localization at high density, consistent with them continuing in cycle instead of entering quiescence.

Additionally, we stained the cells for phospho-Rb (Ser807/811). The retinoblastoma protein is a central regulator of the cell cycle, and becomes dephosphorylated in G0, but is progressively phosphorylated and inactivated as cells leave G0 and move through G1 into S phase (Figure 8D). Rb phosphorylation was very low in NT control at high density but loss of TRAF3 induced a dramatic increase in phospho-Rb positive cells (new Figure 8E, F). Together, these data strongly argue that the key effect of loss of TRAF3 is a failure to enter G0 under conditions that normally promote quiescence.

3) Even though TRAF3 KO relieves cell density control, it remains unclear how the system would be regulated under physiological conditions. Is TRAF3 expression induced at high density? How are the TRAF3 interactors TRAF2 and cIAP1/2 regulated? Does cIAP inhibition by a SMAC-mimetic exert the same effect as TRAF3 KO? SMAC-mimetics deplete cIAP1/2 rapidly and thus effects could be determined at low density or after epithelial cells have been grown to high density.

We have not examined *TRAF2* or cIAP1/2. We did perform western blots for canonical and non-canonical NF-κB components at high and low density. We observed that levels of non-canonical NF-κB signaling proteins are independent of density. Immunoblotting for the canonical NF-κB pathway components showed activation of IKKα/β phosphorylation at high density both in control and TRAF3 KO cells; but no increase in p65 phosphorylation or IκB degradation was observed in control and TRAF3 KO cells either at low or at high density.

We have also added data on the effect of cIAP inhibition by a SMAC-mimetic BV6 at low and high density. BV6 caused over-proliferation of EpH4 cells at high density but not at low density (New Figure 5H, I).

Specific points:The short summary on the putative functions of TRAF3 is rather incomprehensive and not very helpful here.

We have removed this section of the text.

Figure 3: What is the effect of an NF2 KO in primary mammary organoids?

We have not focused on NF2 in our study because NF2 was previously shown to inhibit cell overgrowth through its regulation of HIPPO signaling. Therefore, we do not believe that knocking out NF2 in the organoids would be relevant to this study.

Figure 4: It is well established that TRAF3 is primarily regulating non-canonical NF-κB signaling, but this may exert secondary effects on canonical NF-κB. Actually, the enhanced expression of IkBa indicates that the canonical NF-κB inhibitor may be transcriptionally induced by RelB/p52. As shown for p100/p52 in Figure 4G, the authors should provide a more comprehensive analyses of NFkB/IkB family members (e.g. p65, RelB, p100/p50, IkBa), their activity (p-p65, pIkBa) and some upstream regulators (such as IKKa/b, TRAF3, TRAF2 – see also main point 3), to get a conclusive picture on the impact of TRAF3 loss under low and high density.

We agree but do not detect any activation of canonical NF-κB signaling, as we show in Figure 4 G. We have now also added Western blots for NF-κB components at high and low cell density (Figure 4D, E, H).

Figure 6. I think by sorting mCherry G0/G1 phase for transcriptomic analyses, the authors may have unintentionally sorted out the relevant candidate genes from their analyses, such as D- or E-type cyclins, which are induced for G1-S phase transition.

Because cells lacking Traf3 are more likely to enter the cell cycle at high density, then we would fully expect there to be all the expected changes in gene expression associated with progression through S/G2/M phases. This does not, however, explain why the cells are more likely to enter the cell cycle. We predict that loss of TRAF3 must induce some constitutive change in gene expression in all the cells in the culture, which increases the probability of cell cycle entry (or escape from G0 into G1). We have now testing whether escape from G0 is increased by staining for phosphorylated Rb and using a live cell CDK activity biosensor, as described above.

Reviewer #2 (Significance (Required)):The methodology of CRISPR/Cas9 screening for density-dependent cell cycle arrest is very interesting, powerful and provides a strong advance, because it may be utilized under different conditions. However, it is less clear whether the identified candidate TRAF3 is indeed solely involved in cell density control or more generally restricting proliferation of epithelial cells. Further, the lack of functional data connecting TRAF3 and non-canonical NF-κB to density control is a drawback. Thus, in its current form the study provides mainly a methodological advance. I think more functional data are needed to also provide a conceptual advance and strong visibility for the audience interested in cell cycle and NF-κB regulation in epithelial cells and cancer.Expertise: Signal transduction, immune response, oncogenic signaling in cancerReviewers cross commenting:I agree with rev#1 that “epithelial homeostasis” may be overstated, especiallysince it is not clear, if the TRAF3-NIK-p100 axis may not promote proliferation in other cells as well. Thus, I think adding other cell lines (also non-epithelial) (see rev#1) and providing data, if TRAF3-NIK-p100 is more generally regulating cell cycle or selectively density-controlled proliferation (my comments) is important. Other more mechanistic studies such as Trp53 status, TRAF3 and cyclinD/E regulation and effects of SMAC-mimetics can be done in parallel and results will certainly improve the understanding how loss of TRAF3 influences cell-density control.